# REFINESTAT: EFFICIENT EXPLORATION FOR PROBABILISTIC PROGRAM SYNTHESIS

**Madhav Kanda**    **Shubham Ugare**    **Sasa Misailovic**
University of Illinois Urbana–Champaign
{madhav3, sugare2, misailo}@illinois.edu

## ABSTRACT

Probabilistic programming offers a powerful framework for modeling uncertainty, yet statistical model discovery in this domain entails navigating an immense search space under strict domain-specific constraints. When small language models are tasked with generating probabilistic programs, they frequently produce outputs that suffer from both syntactic, and semantic errors, such as flawed inference constructs. Motivated by probabilistic programmers' domain expertise and debugging strategies, we introduce REFINESTAT, a language model–driven framework that enforces semantic constraints ensuring synthesized programs contain valid distributions, well-formed parameters, and then applies diagnostic-aware refinement by resampling prior or likelihood components whenever reliability checks fail. We evaluate REFINESTAT on multiple probabilistic-programming code-generation tasks using smaller language models (SLMs) and find that it produces programs that are both syntactically sound and statistically reliable, often matching or surpassing those from closed-source large language models (e.g., OpenAI o3). Our code is available at https://github.com/structuredllm/RefineStat.

## 1 INTRODUCTION

Scientific discovery often requires expressing complex systems as statistical models. Finding appropriate models that are both interpretable and computationally efficient is challenging. The vision of automating model discovery has a long-standing history. Past approaches have demonstrated success across various domains, such as identifying physical laws (Bongard & Lipson, 2007; McKinney et al., 2006; Linka et al., 2023), recovering the structure of nonlinear dynamical systems (Schmidt & Lipson, 2009), performing structure-aware nonparametric regression (Duvenaud et al., 2013), and tackling unsupervised learning problems (Grosse, 2014). However, they typically relied on significant manual effort – experts were required to define a domain-specific language (DSL) for representing models and engineer custom search algorithms for exploring compositions within that DSL.

Large Language Models (LLMs) have the potential to automate the model discovery by leveraging their extensive knowledge across various domains, enabling them to propose modeling approaches that were traditionally developed by human experts. However, using LLMs comes with significant challenges. Directly querying LLMs to generate statistical models often produces semantically flawed and unreliable programs, particularly in probabilistic programming languages like PyMC and NumPyro that evolve rapidly. These bugs hinder the correct execution of programs and constrain the effective exploration of the search space of working solutions. Further, running LLMs is costly, as expressive models (e.g., GPT-4) incur high API fees.

These limitations motivate Small Language Models (SLMs) as a practical alternative. Despite recent gains on coding tasks, they still produce *semantic* bugs code that runs but violates the intended statistical meaning. For example, in PyMC (Salvatier et al., 2016), an SLM can generate code that places variance where a standard deviation is expected – `pm.Normal(..., sigma=sigma**2)` instead of `pm.Normal(..., sigma=sigma)`. This change encodes the wrong statistical model, often inflating uncertainty and even triggering errors such as `SamplingError`. In addition, SLMs may produce other semantic mistakes, such as using an invalid argument name `sd` in place of `sigma`, which raises a `TypeError` at model construction time. These cases illustrate the need for our constraints and Bayesian-workflow checks (Gelman et al., 2020) to ensure correctness.

**Our Work: REFINESTAT** We present REFINESTAT, a novel probabilistic programming synthesis framework that efficiently guides a language model to generate probabilistic programs. REFINESTAT is the first to demonstrate that open-weight SLMs can synthesize *reliable* probabilistic programs in the Bayesian-workflow sense i.e., they satisfy standard checks, such as adequate effective sample size, low number of MCMC divergences and strong out-of-sample fit (Section 2 presents the full list).

REFINESTAT produces reliable statistical programs through a two-phase approach: (1) semantically constrained generation and (2) diagnostic-aware refinement (Section 3). In this context, semantically constrained denotes adherence to programming-language semantics (e.g., distribution validity, parameter consistency, proper data types), rather than the linguistic semantics of natural languages. Our semantic constraining ensures that synthesized probabilistic programs contain valid distributions with well-formed parameters, proper variable dependencies, and adherence to PyMC semantics. The diagnostic-aware refinement systematically resamples prior specifications or likelihood models when generated programs fail to meet established reliability criteria within the Bayesian workflow, thereby ensuring efficient search of probabilistic models using small language models.

We evaluate REFINESTAT on a suite of five representative probabilistic datasets, and five open-weight LLMs, with up to 8 billion weights. Our comparison shows that REFINESTAT significantly improves over directly querying LLMs in an unconstrained manner or only syntactic constraining with Syncode (Ugare et al., 2024). We show that the programs generated by REFINESTAT often pass the diagnostic metrics that indicate high quality to represent and explain the data. We also show that the REFINESTAT's performance is comparable to a recent LLM-based generation algorithm BoxLM (Li et al., 2024), which uses two GPT-4 LLM instances to iteratively propose a likely program and refine it, respectively; yet REFINESTAT obtains those results with a single small language model.

**Contributions:** The main contributions of this paper are:

- **Approach:** We present REFINESTAT, a novel SLM-based framework for synthesis of probabilistic programs that are semantically correct and have high predictive performance.
- **Constrained decoding:** We propose using semantic constrained decoding to help generate syntactically and semantically valid probabilistic programs, at a small overall cost.
- **Iterative program search:** We present an iterative refinement loop that leverages a single, unmodified open-weights SLM to generate probabilistic programs with improved diagnostic metrics, refining statistical reliability by selectively resampling the likelihood and prior.
- **Evaluation:** We demonstrate that REFINESTAT performs significantly better than baseline language models, in terms of different diagnostic metrics, and in some cases performs equally well as GPT-4 and hand-written developer programs.

## 2 BACKGROUND

**Language Models.** Current autoregressive language models (LMs) operate on a vocabulary $V \subseteq \Sigma^*$ of tokens. A tokenizer converts an input prompt $O_0 \in \Sigma^*$ into a sequence of tokens $t_1, t_2, \ldots, t_k$. The LM $M : V^* \to \mathbb{R}^{|V|}$ takes this sequence and outputs scores $\mathcal{S}$ over the vocabulary: $\mathcal{S} = M(t_1, t_2, \ldots, t_k)$. A softmax function transforms these scores into a probability distribution, from which $t_{k+1}$ is sampled. Appendix A.1 has more details on decoding and grammar-guided generation.

**Bayesian Workflow.** A robust Bayesian analysis follows an iterative workflow of model specification, posterior inference, diagnostic checking, and model comparison (Gelman et al., 2020). This process can be summarized as: (1) specify the model (likelihood and priors, in our case using an LLM), (2) perform posterior inference, (3) conduct posterior predictive checks and convergence diagnostics, (4) if diagnostics pass, estimate out-of-sample fit (i.e., how well the model would predict data not used in fitting), and (5) compare and rank models by their relative out-of-sample performance (with uncertainty). Further details about diagnostics and predictive evaluation in Appendix A.2.

**Probabilistic Programming.** Statistical modeling aims to describe relationships between variables in data through joint probability distributions that capture both observed phenomena and underlying latent structure. In probabilistic modeling, we formalize this as a joint distribution $p(x, z|\eta)$, where $x = x_{1:n}$ represents $n$ observed data points, $z = z_{1:m}$ denotes $m$ latent variables, and $\eta$ corresponds to fixed model parameters. The inferential goal is to compute the posterior distribution $p(z|x)$, which quantifies uncertainty in the latent variables conditional on observed data. Probabilistic programming languages (PPL) provide a flexible computational substrate for specifying joint distributions $p(x, z \mid \eta)$

as programs while leveraging automated inference methods (e.g., MCMC, variational inference) to compute the posterior $p(z \mid x)$ (van de Meent et al., 2021). Further details in Appendix A.3.

**Probabilistic Programming Diagnostics** We briefly define the standard probabilistic programming diagnostics and metrics used in our framework. Detailed formal definitions are in Appendix B:

1. $\widehat{R}_\phi$: Split-$\widehat{R}$ statistic for parameter $\phi$, measuring MCMC chain convergence.
2. $\mathrm{ESS}_{\mathrm{bulk},\phi}$: The effective bulk sample size for the parameter $\phi$, estimating the sampling efficiency across the central mass of the posterior.
3. $\mathrm{ESS}_{\mathrm{tail},\phi}$: Tail effective sample size for parameter $\phi$, measuring sampling efficiency in tails.
4. Divergences($M$): Count of divergent NUTS (Hoffman et al., 2014) transitions in model $M$.
5. BFMI($M$): Bayesian Fraction of Missing Information for model $M$, assessing energy transition efficiency in Hamiltonian Monte Carlo (HMC) (Neal et al., 2011) algorithm.
6. $\widehat{k}_i(M)$: Pareto shape parameter for observation $i$ in PSIS-LOO (Pareto-smoothed importance sampling leave-one-out cross-validation), quantifying reliability of importance sampling estimates. PSIS-LOO approximates exact LOO predictive densities by smoothing raw importance weights with a generalized Pareto fit to stabilize high-variance weights (Definition 1 below).
7. $\widehat{\mathrm{elpd}}$: Expected Log Pointwise Predictive Density under Leave-One-Out cross-validation, measuring model's out-of-sample predictive accuracy.

To evaluate out-of-sample predictive accuracy, we rely on the expected log pointwise predictive density under leave-one-out cross-validation (ELPD-LOO). A direct computation of LOO requires refitting the model $n$ times (once for each observation), which is often too costly in practice. To avoid this, we use Pareto-smoothed importance sampling leave-one-out cross-validation (PSIS-LOO) (Vehtari et al., 2017), which provides a fast approximation to ELPD-LOO and also gives diagnostics on influential observations via the Pareto $\widehat{k}$ values:

**Definition 1 (PSIS-LOO (Vehtari et al., 2017))** *Let $\mathcal{D} = \{y_i\}_{i=1}^n$ be the observed data and $M$ a model. Draw $\{\theta^{(s)}\}_{s=1}^S$ from $p(\theta \mid \mathcal{D})$, compute raw importance weights $w_i^{(s)} = 1/p(y_i \mid \theta^{(s)})$, and let $\tilde{w}_i^{(s)}$ denote the Pareto-smoothed weights, obtained by replacing the largest tail weights with a generalized Pareto fit. Then the PSIS-LOO estimate is $\widehat{\mathrm{elpd}}_{\mathrm{PSIS\text{-}LOO}} = \sum_{i=1}^n \log\left[\frac{1}{S}\sum_{s=1}^S \tilde{w}_i^{(s)} p(y_i \mid \theta^{(s)})\right]$.*

Instead of refitting the model $n$ times, PSIS-LOO relies on a single full-data fit and stabilized importance weights. The generalized Pareto fit yields shape parameters $k_i$, which assess the reliability of the approximation; following standard guidance, the estimate is considered unreliable if about 20% of the $k_i$ exceed 0.7.

## 3 REFINESTAT

Figure 1 presents REFINESTAT's two main ideas. First, we prune the search space of possible probabilistic programs by enforcing semantic validity during generation, mapping validation rules to nodes

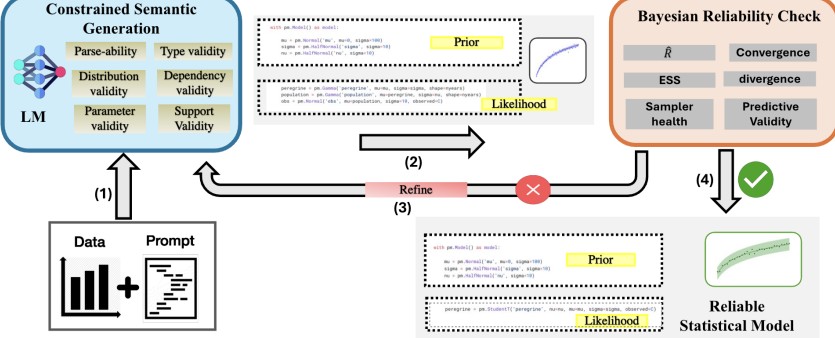

Figure 1: REFINESTAT workflow: (1) A user provides data and prompt to the language model, which generates a probabilistic program. (2) Constrained semantic decoding enforces syntactic and semantic validity of the generated program. (3) A Bayesian reliability check diagnoses convergence, divergences, and predictive validity. If failures are detected, the model is refined by backtracking and resampling priors or likelihoods. (4) Upon passing checks, we get final reliable probabilistic program.

in the partial parse tree and resampling problematic program fragments when constraints are violated. Second, we implement diagnostic-aware refinement, systematically resampling components of statistically unsound models to satisfy Bayesian Workflow guidelines. This integrated approach aims to improve semantic correctness, statistical reliability, and yield strong predictive performance. An illustrative example is provided in Appendix C.1.

**Problem Statement.** Let $\mathcal{D}$ denote the dataset for a given statistical modeling task. The objective of REFINESTAT is to construct a statistical model $M$ in a probabilistic programming language that provides accurate predictive performance while quantifying uncertainty in a fully Bayesian manner.

Our approach to finding a model $M$ that explains the data will follow the standard Bayesian workflow (Gelman et al., 2020) The key challenge to finding such a model $M$ is to automatically compare various candidate models that an LLM produces. To automate this task, we will use a battery of diagnostics from statistical literature (Section 2), computed during the posterior inference in the standard Bayesian workflow (Gelman et al., 2020).

**Definition 2 (Bayesian Workflow Reliability Score)** *Let $\mathcal{M}$ be the set of candidate models, and fix thresholds $\alpha_R, \beta_{\text{bulk}}, \beta_{\text{tail}}, \gamma, L_{cd}, \epsilon$. For each $M \in \mathcal{M}$, we define seven indicator functions* $s_j(M) \in \{0,1\}$ *by* $\mathbb{I}[A] = \begin{cases} 1, & \text{if event } A \text{ holds,} \\ 0, & \text{otherwise,} \end{cases}$ *and set*

1. $s_1(M) = \mathbb{I}\left[\max_\phi \widehat{R}_\phi(M) \leq \alpha_R\right],$ 
   2. $s_2(M) = \mathbb{I}[\text{BFMI}(M) > \gamma],$

3. $s_3(M) = \mathbb{I}\left[\min_\phi \text{ESS}_{\text{bulk},\phi}(M) \geq \beta_{\text{bulk}}\right],$
   4. $s_4(M) = \mathbb{I}[\text{divergences}(M) = 0],$

5. $s_5(M) = \mathbb{I}\left[\min_\phi \text{ESS}_{\text{tail},\phi}(M) \geq \beta_{\text{tail}}\right],$
   6. $s_6(M) = \mathbb{I}\left[\frac{1}{n}\sum_{i=1}^{n}\mathbb{I}[\widehat{k}_i(M) \leq L_{cd}] \geq 1 - \epsilon\right],$

7. $s_7(M) = \mathbb{I}\left[\widehat{\text{elpd}}(M) \text{ is finite}\right].$
   *Then the* $\boxed{\text{reliability score}}$ *is* $\mathcal{B}(M) = \sum_{j=1}^{7} s_j(M).$

These diagnostics can be extracted directly from the MCMC engines (e.g. Stan (Carpenter et al., 2017a), PyMC (Salvatier et al., 2016)). Although $\widehat{\text{elpd}}$ provides a principled Bayesian measure of out-of-sample predictive accuracy, its Monte Carlo estimate can be unreliable if the sampler has not fully converged or if the importance weights are unstable (Gelman et al., 1995). To mitigate these risks, we consider elpd estimates for models that satisfy standard convergence thresholds, thus ensuring that predictive comparisons rest on reliable posterior samples.

We require each model to pass *most* of these seven checks: if any check fails, the corresponding $s_j(M)$ is zero, and the total score reflects how many diagnostics remain satisfactory. The final check concerns the availability of the $\widehat{\text{elpd}}(M)$ estimate; if $\widehat{\text{elpd}}(M)$ cannot be computed or is infinite, then $s_{\text{ELPD}}(M) = 0$, and the model is treated as failing that diagnostic. A higher overall score indicates that more diagnostics have passed, so *when* $\widehat{\text{elpd}}(M)$ is available, the resulting estimate can be trusted with greater confidence. We consider a model reliable once its score exceeds a cutoff $\zeta$. We use $\zeta = 5$ to allow marginal diagnostic failures while maintaining confidence in the reported $\widehat{\text{elpd}}$.

**Definition 3 (Valid model space)** *For a set of candidate models $\mathcal{M}$, a valid model space is:*
$$\mathcal{M}_{\text{valid}} = \left\{ M \in \mathcal{M} : \mathcal{B}(M) \geq \zeta \right\}.$$

**Definition 4 (REFINESTAT Objective)** *We finally define our objective to identify the model that attains the highest ELPD-LOO estimate within the space of models $\mathcal{M}_{\text{valid}}$:*
$$\boxed{M^* = \arg\max_{M \in \mathcal{M}_{\text{valid}}} \widehat{\text{elpd}}(M),}$$

### 3.1 SEMANTICALLY-CONSTRAINED PROBABILISTIC PROGRAM GENERATION

We formalize the generation of semantically valid probabilistic programs via iterative constrained sampling. Let $G = (\mathcal{N}, \mathcal{T}, \mathcal{P}, S_0)$ be a context-free grammar with nonterminal symbols $\mathcal{N}$, terminal symbols $\mathcal{T}$, production rules $\mathcal{P}$, and start symbol $S_0$.

For a partial program with parse tree $\kappa$, let $\mathcal{F}(\kappa)$ denote the set of program fragments, where each fragment $n \in \mathcal{F}(\kappa)$ is a rooted subtree of $\kappa$ corresponding to a single syntactic statement. *Validation functions* operate on a fragment $n$ within program context $\pi \in \Pi$: $\Phi : \mathcal{F}(\kappa) \times \Pi \to \{0, 1\}$. These functions are conjunctions of individual correctness checks:

$$\Phi(n, \pi) = \phi^1(n, \pi) \land \phi^2(n, \pi) \land \cdots \land \phi^m(n, \pi)$$

Each fragment thus corresponds to a single statement, possibly comprising multiple AST nodes.

**Validity Predicates for Probabilistic Program Fragments.** To ensure the correctness of synthesized probabilistic program fragments, we define three essential validation predicates. Let $\mathcal{F}(s)$ denote the set of all probability distribution functions invoked within fragment $s$. The validation predicates are:

1. **Parse-ability:** $\phi_1(s, \Pi) = 1$ if the fragment conforms to the grammar $G$.
2. **Distribution validity:** $\phi_2(s, \Pi) = \prod_{f \in \mathcal{F}(s)} \mathbf{1}\{f \in \mathcal{M}\}$ verifies that each probabilistic operation $f$ exists in the available library $\mathcal{M}$ of PPL.
3. **Parameter validity:** $\phi_3(s, \Pi) = \prod_{f \in \mathcal{F}(s)} \mathbf{1}\{P(f) \subseteq P_{\text{acc}}(f)\}$ confirms that operation parameters $P(f)$ adhere to the accepted specifications $P_{\text{acc}}(f)$, essential for maintaining probabilistic semantics. i.e. we ensure that the provided parameters for any distribution are correct according to the distribution's specification. For instance, Figure 2 shows parameter "sd" was invalid and resampled correctly as "sigma".
4. **Dependency validity:** $\phi_4(s, \Pi) = \prod_{v \in \text{Vars}(s)} \mathbf{1}\{\text{all dependencies of } v \text{ are defined before use}\}$ ensures that random variables are declared and initialized before they are referenced.
5. **Support validity:** $\phi_5(s, \Pi) = \prod_{f \in \mathcal{F}(s)} \mathbf{1}\{P(f) \in \text{Supp}(f)\}$ confirms that parameter values fall within the distribution's support (e.g., variance $> 0$, probabilities in $[0, 1]$).
6. **Type validity:** $\phi_6(s, \Pi) = \prod_{f \in \mathcal{F}(s)} \mathbf{1}\{\text{type}(P(f)) \in T(f)\}$ checks that each parameter $P(f)$ has the expected type from the specification $T(f)$, e.g., ensuring numeric values for scale parameters, or integer values for counts.

The final predicate $\Phi(s, \Pi) = \bigwedge_{i=1}^{6} \phi_i(s, \Pi)$ ensures that generated fragments satisfy all requirements of the probabilistic programming language (1,4,6) and the Bayesian model (2,3,5). Our generation algorithm leverages these properties by maintaining a global symbol table $\Pi : \mathcal{A} \to \mathcal{M}$ mapping each alias $a \in \mathcal{A}$ to its module or namespace $m \in \mathcal{M}$.

Generation proceeds via local rejection sampling on $S_N$: we repeatedly sample $s \sim S_N$ until finding $s^*$ where $\Phi(s^*, \Pi) = 1$. This iterative process continues until a termination fragment is generated, ensuring every component in the final probabilistic program satisfies all semantic constraints. Our local rejection sampling is token-efficient and we backtrack and precisely resample the tokens that correspond to the vio-

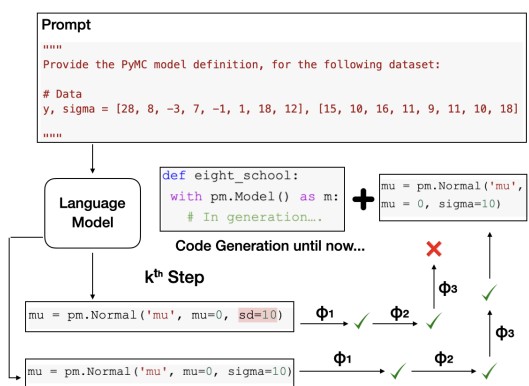

Figure 2: Constrained decoding in REFINESTAT fixing a `TypeError` from using `sd` instead of `sigma`, as illustrated in Section 1.

lation of the constraints. The approach is particularly effective for languages like PyMC, where maintaining consistent probabilistic variable scopes and dependencies is critical. Combining syntactic constraints with semantic validation enables efficient exploration of the program space while ensuring the probabilistic soundness of generated models.

### 3.2 Program Validation and Guided Resampling

Building on the validation predicate $\Phi$, we now formalize our constrained generation and refinement methodology. We introduce the *constrained-decoding operator* $\mathcal{L}_{\text{CD}} : \mathcal{C} \to \mathcal{B}$, which implements our validation-guided sampling. Here, a *statement* is an individual syntactic unit, and a *code block* $\mathcal{B}$ is a (possibly multi-statement) sequence of such statements. For any context $\mathcal{C}$, this operator returns a new code block $\mathcal{B}$ satisfying $\Phi(\mathcal{C} \| \mathcal{B}) = 1$, where $\|$ denotes sequential concatenation of blocks;

concretely, for two blocks $A$ and $B$, $A\|B$ is the program text formed by appending all statements of $B$ immediately after those of $A$.

$\mathcal{D}\|\mathcal{P}\|\mathcal{L}$ denotes the full probabilistic program with data $\mathcal{D}$, prior $\mathcal{P}$, and likelihood $\mathcal{L}$. During refinement we perform *resampling* via two steps: $\mathcal{L} \leftarrow L_{cd}(\mathcal{D}\|\mathcal{P})$ (likelihood resampling), $\mathcal{P} \leftarrow L_{cd}(\mathcal{D})$ (prior resampling), guaranteeing replaced blocks remain semantically valid.

Before refinement begins, the data block $\mathcal{D}$ is taken directly from the user prompt, while the initial prior and likelihood blocks are generated via constrained decoding, i.e., $\mathcal{P} \leftarrow \mathcal{L}_{\mathrm{CD}}(\mathcal{D})$ and $\mathcal{L} \leftarrow \mathcal{L}_{\mathrm{CD}}(\mathcal{D} \| \mathcal{P})$. Algorithm 1 synthesizes programs in two phases. First, it checks semantic correctness via $\Phi$, ensuring parseability, distribution validity, and parameter consistency. Second, it evaluates Bayesian diagnostics $d_1, \ldots, d_L$ on the full program $\mathcal{D}\|\mathcal{P}\|\mathcal{L}$; at least $K$ thresholds $\{\tau_j\}$ must be met to accept a candidate. As shown in Figure 1, if diagnostics

**Algorithm 1** REFINESTAT Synthesis via $\mathcal{D}\|\mathcal{P}\|\mathcal{L}$

---
**Require:** $R_{\max}$, $\alpha$, $\beta$, $\{\tau_j\}_{j=1}^{L}$, $K$
1: $r \leftarrow 0$, $\ell \leftarrow 0$, $\mathcal{V} \leftarrow \varnothing$, $\mathcal{P} \leftarrow \varnothing, \mathcal{L} \leftarrow \varnothing$
2: **while** $r < R_{\max}$ **and** $|\mathcal{V}| < \beta$ **do**
3:     $\mathrm{Prog} \leftarrow \mathcal{D}\|\mathcal{P}\|\mathcal{L}$
4:     **if** $\neg\Phi(\mathrm{Prog})$ **then**
5:         $r \leftarrow r + 1$; **continue**
6:     compute diagnostics $d_1, ..., d_L$ on Prog
7:     $P_{\mathrm{pass}} \leftarrow \big|\{j : d_j \geq \tau_j\}\big|$
8:     **if** $P_{\mathrm{pass}} \geq K$ **then**
9:         $\mathcal{V} \leftarrow \mathcal{V} \cup \{\mathrm{Prog}\}$; **continue**
10:     **if** $\ell < \alpha$ **then**    ▷ likelihood resampling
11:         $\mathcal{L} \leftarrow L_{cd}(\mathcal{D}\|\mathcal{P})$
12:         $\ell \leftarrow \ell + 1$
13:     **else**              ▷ prior resampling
14:         $\mathcal{P} \leftarrow L_{cd}(\mathcal{D})$
15:     $r \leftarrow r + 1$
16: **return** $\arg\max \mathrm{elpd}(M)$ over all $M \in \mathcal{V}$

---

fail, we perform one of two *resampling* steps to refine the program: (i) likelihood resampling replaces $\mathcal{L}$ under the data–prior context, addressing convergence or sampler-health issues; (ii) prior resampling replaces $\mathcal{P}$ under the data context, correcting prior-specification errors. We iterate until we collect $\beta$ valid programs or exhaust the budget $R_{\max}$, and return the program maximizing $\widehat{\mathrm{elpd}}$.

## 4 EXPERIMENTAL METHODOLOGY

We use PyMC, a Python probabilistic programming library, to perform inference. We provide the same initial prompt while performing unconstrained generation and using REFINESTAT. We prompt the model by providing it with the dataset, the necessary library and the text query. The exact format of the prompt is provided in the Appendix D. As stated in Definition 2, we assess model reliability using standard Bayesian diagnostics (Vehtari et al., 2021; 2017; Gelman et al., 1995). Further details on hyperparameters and experimental setup are provided in Appendix E

**Datasets.** We use five benchmark datasets from Stan PosteriorDB (Magnusson et al., 2024), mirroring the selection in prior research on automated statistical modeling (Li et al., 2024):

- **Eight Schools (Rubin, 1981):** This dataset originates from a study commissioned by the Educational Testing Service, which examines the effects of coaching programs on test performance.
- **Dugongs (Unit, b):** This dataset provides measurements on the ages and lengths of 27 dugongs.
- **Surgical (Unit, a):** This dataset comprises records on the number of cardiac surgeries performed on infants, along with the associated failure rates.
- **Peregrine (M Kery, 2011):** This dataset tracks the breeding trajectory of the peregrine falcon population in the French Jura region from 1964 to 2003.
- **GP:** This dataset contains simulated observations generated from a Poisson Gaussian Process.

**Models.** We experiment with a range of state-of-the-art LLMs, spanning multiple parameter scales including Qwen2.5 (code-specific) (Hui et al., 2024), models from the Llama series, DeepSeek, and Google's CodeGemma. We have used a total of four models including, Llama3-8B (Grattafiori et al., 2024), CodeGemma-7B (Team et al., 2024), Qwen2.5-Coder-7B (Hui et al., 2024), and DeepSeek-R1-Distill-Qwen-7B (hereafter we refer to it as "DQ-7B") (Guo et al., 2025).

## 5 EXPERIMENTAL RESULTS

### 5.1 IMPROVED RUN RATE OVER UNCONSTRAINED AND SYNTAX-DRIVEN GENERATION

We conduct a comprehensive evaluation on diverse datasets, comparing REFINESTAT against both an unconstrained baseline and leading language models across a suite of diagnostic and performance

metrics. We used identical prompts across our framework, the Standard baseline (unconstrained), syntactically-constrained tool Syncode (Ugare et al., 2024), and REFINESTAT.

Table 1 presents Run rates across different temperature settings. Run rate is the fraction of programs that successfully produce the samples from the posterior distribution. The problems that do not run successfully include those with runtime errors such as (1) numerical errors, e.g., inf/nan, (2) sampling issues due to unlikely prior parameterization, (3) other sampling warnings, e.g., failed to initialize chains, (4) static compilation issues.

REFINESTAT achieves success rates approximately 40 percentage points higher than the Standard baseline and 30 points higher than Syncode, demonstrating that our validation-guided approach substantially enhances code generation reliability by mitigating both syntactic and semantic error sources. These results show that

Table 1: Run rates for Standard, SYNCODE, and REFINESTAT by temperature

| Temp. | Standard | SYNCODE | REFINESTAT |
|-------|----------|---------|------------|
| 0.2 | 0.10 | 0.21 | **0.45** |
| 0.3 | 0.11 | 0.21 | **0.50** |
| 0.4 | 0.11 | 0.21 | **0.50** |

REFINESTAT significantly enhances code generation reliability. The top root causes of failure are syntax errors, semantic errors, and sampler pathologies. We categorize these failures and provide a detailed discussion in Appendix F.1.

## 5.2 COMPARISON OF GENERATED PROGRAM QUALITY TO UNCONSTRAINED BASELINE

To evaluate semantic correctness and diagnostic robustness, we compared programs generated by the base language model with those from REFINESTAT under identical prompts. Since we observed that REFINESTAT consumes almost twice the number of tokens used by the Baseline (see Appendix G.1), we run baseline models five times with different seeds ($2.5\times$ tokens more than REFINESTAT) and compare the best program based on Bayesian Reliability Score and ELPD LOO to a single REFINESTAT run. We repeat this process five times to compute the mean and standard deviation for all metrics. In Table 2, we report five representative metrics drawn from these diagnostics (for space reasons). Bold entries denote cases where REFINESTAT outperforms the corresponding baseline. Since the Reliability Score aggregates multiple diagnostic metrics, it is highlighted most frequently; when Reliability Scores are similar, we instead emphasize differences in ELPD LOO. The symbol ✗ indicates that no valid program was produced—i.e., the method failed to explore the search space sufficiently to yield a correct result.

Except for one instance, REFINESTAT matches or exceeds the Standard Baseline in terms of the Bayesian Workflow Reliability score, and in some cases achieves up to twice the reliability. Moreover, REFINESTAT delivers substantially better performance on individual diagnostics, particularly divergences. Notably, DQ-7B, which failed on every dataset under the Standard Baseline, succeeded on all datasets when augmented with REFINESTAT. For example, on the Surgical dataset with Meta-Llama, the Standard Baseline produces over 1000 divergences, while REFINESTAT produces none.

We observed that in cases where the Standard baseline attains a higher ELPD-LOO than REFINESTAT, closer inspection of the diagnostics exposes unreliable sampling. For instance, on the Meta-Llama GP task the Standard model achieves a superior ELPD score, but exhibits split-$\widehat{R} = 4.13 \gg 1$ and a low reliability score (3), indicating severe convergence issues. In contrast, REFINESTAT may report a marginally lower ELPD yet maintains $\widehat{R} \approx 1$ and a higher reliability score, reflecting trustworthy posterior estimates. Furthermore, REFINESTAT delivers markedly lower variability in key diagnostics such as $\widehat{R}$ and the number of divergent transitions demonstrating its consistency and robustness. We further illustrate the types of structural changes introduced during refinement in Appendix C.2.

## 5.3 COMPARISON OF GENERATED PROGRAM QUALITY TO BOXLM

Table 3: Comparison of ELPD LOO scores with BoxLM (Li et al., 2024), REFINESTAT using DQ-7B, and Expert values

| Dataset | Expert | REFINESTAT w/ DQ-7B(mean ± std) | BoxLM w/ GPT-4 | OpenAI-o3 (mean ± std) |
|---------|--------|-------------------------------|----------------|------------------------|
| Eight schools | -30.70 | -30.68 ± 0.11 | -30.42 | -30.74 ± 0.07 |
| Dugongs | 22.43 | 8.35 ± 0.06 | 23.40 | 22.83 ± 8.12 |
| Peregrine | -112.60 | -114.29 ± 2.76 | -173.11 | -133.29 ± 10.33 |
| Surgical | -39.73 | -46.51 ± 3.49 | -38.03 | -38.73 ± 0.51 |
| GP | -26.53 | -23.39 ± 1.14 | – | -34.95 ± 13.28 |

We compare REFINESTAT's performance using the DQ-7B model (averaged over five runs) against three baselines: the *Expert* stan programs from PosteriorDB (Magnusson et al., 2024), the BoxLM system introduced by Li et al. (2024), and programs generated by OpenAI o3. Since the code for BoxLM is not publicly available, we

Table 2: Comparison of Diagnostic Scores and ELPD-LOO for Standard vs. REFINESTAT

| Dataset | Model | Variant | Reliab. Scr.↑ Mean | Std | $\widehat{R}$↓ Mean | Std | ESS Bulk↑ Mean | Std | Diverg.↓ Mean | Std | Pareto $k$↓ Mean | Std | ELPD LOO↑ Mean | Std |
|---|---|---|---|---|---|---|---|---|---|---|---|---|---|---|
| 8 Schools | Meta-LLama-3-8B | Standard | 7.00 | 0.00 | 1.00 | 0.00 | 2261.00 | 0.00 | 0.00 | 0.00 | 0.00 | 0.00 | -31.70 | 0.00 |
| | | RefineStat | 7.00 | 0.00 | 1.00 | 0.00 | 2303.00 | 768.76 | 0.00 | 0.00 | 0.00 | 0.05 | -31.77 | 0.61 |
| | CodeGemma-7B | Standard | ✗ | ✗ | ✗ | ✗ | ✗ | ✗ | ✗ | ✗ | ✗ | ✗ | ✗ | ✗ |
| | | RefineStat | **7.00** | 0.49 | 1.00 | 0.00 | 2573.00 | 527.68 | 0.00 | 1.97 | 0.00 | 0.06 | -31.46 | 1.84 |
| | Qwen-Coder-7B | Standard | 3.80 | 1.30 | 1.02 | 0.01 | 223.25 | 80.43 | 92.75 | 31.38 | 0.22 | 0.21 | -31.31 | 0.68 |
| | | RefineStat | **5.00** | 0.45 | 1.01 | 0.01 | 256.50 | 883.83 | 34.50 | 86.89 | 0.00 | 0.05 | **-30.80** | 0.05 |
| | DQ-7B | Standard | ✗ | ✗ | ✗ | ✗ | ✗ | ✗ | ✗ | ✗ | ✗ | ✗ | ✗ | ✗ |
| | | RefineStat | **5.00** | 0.98 | 1.02 | 0.02 | 219.00 | 2176.00 | 102.00 | 75.36 | 0.00 | 0.00 | **-30.68** | 0.11 |
| Dugongs | Meta-LLama-3-8B | Standard | 7.00 | 0.00 | 1.00 | 0.00 | 2167.67 | 884.79 | 0.00 | 0.00 | 0.01 | 0.02 | -7.66 | 30.39 |
| | | RefineStat | 7.00 | 0.00 | 1.00 | 0.00 | 1696.00 | 284.33 | 0.00 | 0.00 | 0.00 | 0.02 | **8.42** | 24.51 |
| | CodeGemma-7B | Standard | 5.70 | 2.30 | 1.04 | 0.06 | 1527.33 | 1325.54 | 285.67 | 494.79 | 0.00 | 0.00 | 3.90 | 7.71 |
| | | RefineStat | **7.00** | 0.00 | 1.00 | 0.00 | 1908.00 | 2066.23 | 0.00 | 0.00 | 0.04 | 0.02 | **8.07** | 15.42 |
| | Qwen-Coder-7B | Standard | 7.00 | 0.00 | 1.00 | 0.01 | 1788.67 | 123.43 | 0.00 | 0.00 | 0.04 | 0.00 | 8.15 | 0.29 |
| | | RefineStat | 7.00 | 0.00 | 1.00 | 0.00 | 1683.00 | 148.16 | 0.00 | 0.00 | 0.04 | 0.02 | **8.29** | 0.05 |
| | DQ-7B | Standard | ✗ | ✗ | ✗ | ✗ | ✗ | ✗ | ✗ | ✗ | ✗ | ✗ | ✗ | ✗ |
| | | RefineStat | **7.00** | 0.00 | 1.00 | 0.00 | 2376.00 | 149.61 | 0.00 | 0.00 | 0.00 | 0.03 | 8.35 | 0.06 |
| Peregrine | Meta-LLama-3-8B | Standard | 6.00 | 0.00 | 1.00 | 0.00 | 3774.00 | 0.00 | 7.00 | 0.00 | 0.00 | 0.00 | -184.96 | 0.00 |
| | | RefineStat | **7.00** | 0.00 | 1.00 | 0.00 | 3574.00 | 428.26 | 0.00 | 0.00 | 0.00 | 0.00 | **-173.00** | 4.91 |
| | CodeGemma-7B | Standard | 7.00 | 0.00 | 1.00 | 0.00 | 4261.00 | 0.00 | 0.00 | 0.00 | 0.00 | 0.00 | -172.91 | 0.00 |
| | | RefineStat | 6.50 | 0.53 | 1.00 | 0.00 | 2930.00 | 1343.79 | 0.50 | 0.53 | 0.00 | 0.00 | **-129.93** | 3.91 |
| | Qwen-Coder-7B | Standard | 7.00 | 0.00 | 1.00 | 0.00 | 4238.00 | 0.00 | 0.00 | 0.00 | 0.00 | 0.00 | -173.11 | 0.00 |
| | | RefineStat | 7.00 | 0.00 | 1.00 | 0.00 | 4679.00 | 88.73 | 0.00 | 0.00 | 0.00 | 0.00 | **-172.98** | 0.12 |
| | DQ-7B | Standard | ✗ | ✗ | ✗ | ✗ | ✗ | ✗ | ✗ | ✗ | ✗ | ✗ | ✗ | ✗ |
| | | RefineStat | **6.50** | 0.53 | 1.01 | 0.01 | 963.50 | 436.70 | 25.50 | 27.26 | 0.00 | 0.00 | -114.29 | 2.76 |
| Surgical | Meta-LLama-3-8B | Standard | 5.50 | 1.30 | 1.01 | 0.01 | 1159.75 | 1202.11 | 1066.50 | 1267.61 | 0.65 | 0.44 | -31.59 | 20.02 |
| | | RefineStat | **6.00** | 0.77 | 1.00 | 0.00 | 579.00 | 1272.12 | **0.00** | 5.75 | 0.00 | 0.36 | -46.73 | 36.63 |
| | CodeGemma-7B | Standard | 5.50 | 0.70 | 1.00 | 0.00 | 1230.00 | 503.46 | 6.00 | 5.66 | 0.46 | 0.65 | -42.02 | 5.75 |
| | | RefineStat | **6.00** | 0.49 | 1.00 | 0.00 | 2026.00 | 450.46 | 0.00 | 0.49 | 0.00 | 0.12 | -45.55 | 381.60 |
| | Qwen-Coder-7B | Standard | 6.30 | 1.50 | 1.01 | 0.02 | 1332.75 | 784.13 | 37.50 | 75.00 | 0.23 | 0.46 | -44.48 | 4.35 |
| | | RefineStat | **7.00** | 0.41 | 1.00 | 0.00 | 1642.00 | 1350.46 | 0.00 | 0.00 | 0.00 | 0.38 | -46.55 | 2.79 |
| | DQ-7B | Standard | ✗ | ✗ | ✗ | ✗ | ✗ | ✗ | ✗ | ✗ | ✗ | ✗ | ✗ | ✗ |
| | | RefineStat | **7.00** | 0.50 | 1.00 | 0.01 | 2800.00 | 1182.68 | 0.00 | 0.00 | 0.00 | 0.37 | -46.51 | 3.49 |
| GP | Meta-LLama-3-8B | Standard | 3.00 | 0.00 | 4.13 | 0.00 | 4.00 | 0.00 | 1634.00 | 0.00 | 0.00 | 0.00 | -21.61 | 0.00 |
| | | RefineStat | **6.00** | 0.49 | 1.00 | 0.00 | 1710.00 | 668.57 | 13.00 | 12.39 | 0.09 | 0.08 | -152.30 | 139.07 |
| | CodeGemma-7B | Standard | 7.00 | 0.00 | 1.00 | 0.00 | 6034.00 | 0.00 | 0.00 | 0.00 | 0.18 | 0.00 | -154.42 | 0.00 |
| | | RefineStat | 7.00 | 0.98 | 1.00 | 0.00 | 1752.00 | 1004.88 | 0.00 | 2.46 | 0.00 | 0.49 | **-22.76** | 126.08 |
| | Qwen-Coder-7B | Standard | ✗ | ✗ | ✗ | ✗ | ✗ | ✗ | ✗ | ✗ | ✗ | ✗ | ✗ | ✗ |
| | | RefineStat | ✗ | ✗ | ✗ | ✗ | ✗ | ✗ | ✗ | ✗ | ✗ | ✗ | ✗ | ✗ |
| | DQ-7B | Standard | ✗ | ✗ | ✗ | ✗ | ✗ | ✗ | ✗ | ✗ | ✗ | ✗ | ✗ | ✗ |
| | | RefineStat | **6.50** | 0.53 | 1.01 | 0.00 | 816.50 | 89.27 | 30.50 | 32.61 | 0.00 | 0.00 | -23.39 | 1.14 |

rely on the reported numbers from their paper for comparison. Because the dataset we use for the GP task is not included in BoxLM's evaluation suite, we omit their result for that dataset.

Table 3 presents the ELPD LOO scores, showing that our framework consistently outperforms both BoxLM and OpenAI o3 on the PEREGRINE dataset, and surpasses OpenAI o3 on the GP task, while matching the performance of expert-written programs in most cases. Across the remaining datasets, our approach performs comparably to other baselines, with the exception of DUGONGS, where performance is slightly lower. These results demonstrate that REFINESTAT achieves performance comparable to, and in several cases better than, large language models like OpenAI o3 and multi-agent frameworks such as BoxLM.

## 5.4   ABLATION STUDY

**Effectiveness of Semantic Validation Components.** To evaluate the contribution of each validity predicate within REFINESTAT, we perform an ablation study by systematically disabling one component at a time and measuring the resulting compilation rate. This analysis is conducted across all models with ten different random seeds to ensure robustness. Notably, removing all validation predicate reduces the system to the behavior of SYNCODE.

Table 4 shows that parameter validity emerges as the most critical component, its removal results in a substantial drop of 14.5 percentage points in compilation success. All checks contribute meaningfully to overall performance: omitting distribution validity or parse-ability consistently reduce compilation rates (−9% and −5.5%, respectively), underscoring

Table 4: Ablation Study: Impact of Semantic Validation Checks on Run rate

| # | Method | Run % | ∆Run |
|---|---|---|---|
| 1 | REFINESTAT (all components) | **50.0%** | - |
| 2 | w/o Parameter validity | 35.5% | −14.5% |
| 3 | w/o Distribution validity | 26.5% | −9% |
| 4 | w/o Parse-ability | 21.0% | −5.5% |
| 5 | w/o grammar-guided generation | 11.0% | −10.0 % |

the complementary role these validations play in ensuring soundness of generated programs.

**Memorization Effect.** A number of studies (Dong et al., 2024a; Golchin & Surdeanu, 2025; 2024; Li, 2023) have highlighted concerns about the memorization effect in large language models, where models may reproduce previously seen content rather than demonstrating genuine synthesis. (Kong et al., 2025) addresses this issue by proposing code mutation to reveal potential memorization in program repair tasks. Inspired by this perspective, we introduce dataset and prompt modifications designed to preserve the semantics of the data while changing its presentation. We apply two systematic prompt modifications across all benchmarks when evaluating Meta-Llama 3-8B, with details provided in Appendix G.2. The modified versions match the original REFINESTAT on reliability, convergence, and predictive metrics. Additionally, PosteriorDB provides limited ground-truth PyMC programs: only the Eight Schools model is available, and it targets an outdated version of PyMC, while the remaining programs are provided in Stan. While these studies are limited in scope, and measuring the memorization effect is still an open problem, they give an indication that REFINESTAT's effectiveness may not be the consequence of just memorization.

## 5.5 GENERALIZABILITY ACROSS PROBABILISTIC PROGRAMMING BACKENDS

Although our primary evaluation uses PyMC, the design of REFINESTAT is not specific to any single probabilistic programming library. To show generalizability of the framework, we apply REFINESTAT to NumPyro using the same prompting setup and the Qwen-2.5-Coder–7B model.

Across temperatures, REFINESTAT substantially improves the fraction of programs that successfully compile and execute. Table 5 shows that the run rate more than doubles relative to Standard unconstrained decoding, consistent with the improvements in PyMC3 experiment (Table 1).

Table 6 reports the metrics for NumPyro generated programs across all benchmarks, as those in Table 2 (for PyMC3). For every dataset, REFINESTAT attains equal or higher reliability than the Standard baseline. When both approaches achieve similar reliability, REFINESTAT yields comparable or improved predictive performance. Notably, in the GP task, Standard decoding fails to produce any valid model, whereas REFINESTAT

Table 5: Run-rate comparison when using NumPyro as the inference backend.

| Temp | Standard | REFINESTAT |
|---|---|---|
| 0.2 | 0.17 | **0.34** |
| 0.3 | 0.15 | **0.35** |
| 0.4 | 0.15 | **0.33** |

consistently generates executable programs with stable diagnostics. Overall, these results show that REFINESTAT's semantic filtering and diagnostic-aware refinement generalize across PPL backends, improving program validity and sampling quality beyond PyMC.

Table 6: Diagnostics and ELPD-LOO results for NumPyro using Qwen-2.5-Coder–7B, demonstrating the generalizability of REFINESTAT.

| Dataset | Variant | Reliab. Scr.↑ | | $\widehat{R}$ ↓ | | ESS Bulk ↑ | | Diverg. ↓ | | Pareto $k$ ↓ | | ELPD LOO ↑ | |
|---|---|---|---|---|---|---|---|---|---|---|---|---|---|
| | | Mean | Std | Mean | Std | Mean | Std | Mean | Std | Mean | Std | Mean | Std |
| 8 Schools | Standard | 7.00 | 0.00 | 1.00 | 0.00 | 1769.00 | 0.00 | 0.00 | 0.00 | 0.13 | 0.00 | -31.83 | 0.00 |
| | REFINESTAT | 7.00 | 0.00 | 1.00 | 0.00 | 1850.00 | 50.00 | 0.00 | 0.00 | 0.11 | 0.02 | -31.10 | 0.05 |
| Dugongs | Standard | 7.00 | 0.00 | 1.00 | 0.00 | 1739.00 | 0.00 | 0.00 | 0.00 | 0.04 | 0.00 | 8.18 | 0.00 |
| | REFINESTAT | 7.00 | 0.00 | 1.00 | 0.00 | 1747.00 | 11.55 | 0.00 | 0.00 | 0.04 | 0.00 | 8.19 | 0.01 |
| Peregrine | Standard | 6.00 | 0.00 | 1.00 | 0.00 | 3232.50 | 81.32 | 0.00 | 0.00 | 0.00 | 0.00 | -368.21 | 0.04 |
| | REFINESTAT | 6.67 | 0.47 | 1.00 | 0.00 | 1658.00 | 135.60 | 0.00 | 0.00 | 0.11 | 0.10 | -76.25 | 122.44 |
| Surgical | Standard | 6.00 | 0.00 | 1.00 | 0.00 | 1472.00 | 0.00 | 0.00 | 0.00 | 0.25 | 0.00 | -245.14 | 0.00 |
| | REFINESTAT | 7.00 | 0.00 | 1.00 | 0.00 | 1628.00 | 0.00 | 0.00 | 0.00 | 0.08 | 0.00 | -50.37 | 0.00 |
| GP | Standard | ✗ | ✗ | ✗ | ✗ | ✗ | ✗ | ✗ | ✗ | ✗ | ✗ | ✗ | ✗ |
| | REFINESTAT | 6.67 | 0.47 | 1.00 | 0.00 | 2344.33 | 115.46 | 0.00 | 0.00 | 0.09 | 0.13 | -83.07 | 82.70 |

## 6 RELATED WORK

Researchers have presented probabilistic techniques for discovering model structures from observed data, including Bayesian networks (Mansinghka et al., 2006; Lowd & Domingos, 2012), matrix-composition models (Grosse et al., 2012), Markov networks (Gogate et al., 2010), and deep probabilistic models (Gens & Domingos, 2013). Probabilistic programming languages (PPLs) flexibly represent these models as programs but demand substantial domain and API expertise. Recent advances in LLM code generation make synthesizing probabilistic programs more accessible. REFINESTAT leverages this opportunity to advance the state of the art in LLM-based probabilistic program synthesis.

**Probabilistic Program Synthesis:** There have been many approaches for deterministic program synthesis, which is recently been dominated by LLM-based approaches. Several works (Nori et al., 2015; Saad et al., 2019; Gerasimou et al., 2015; Češka et al., 2019; Ellis et al., 2015) synthesize probabilistic programs using classical machine learning or symbolic methods. Prior work has also proposed techniques for debugging probabilistic programs (Dutta et al., 2019; 2022; Nussbaumer et al., 2026), which could provide further feedback for probabilistic program synthesis. (Gerasimou et al., 2015) uses a genetic algorithm for probabilistic model generation. (Saad et al., 2019) presents techniques to automatically construct probabilistic programs using Bayesian inference over DSLs defined via probabilistic grammars, enabling qualitative structure discovery and quantitative prediction.

Most recently, (Li et al., 2024) uses LLM for probabilistic program synthesis. The paper shows that with instances of GPT4 as the generator and critic (closed LLM) can find reasonable probabilistic programs from data. As our evaluation shows, REFINESTAT significantly improves the ability to find programs that fits the data (recall Table 3) and in contrast to (Li et al., 2024), runs only a single small open LLM ($< 8B$ weights), demonstrating the benefits of constrained decoding.

**Program Synthesis with Constrained LLM Decoding:** Recent advances in program synthesis have enabled constrained decoding approaches where LLMs generate code while adhering to formal language specifications. These constraints can be partially precomputed and enforced more efficiently for regular (Deutsch et al., 2019; Willard & Louf, 2023; Kuchnik et al., 2023) or context-free (Koo et al., 2024; Ugare et al., 2024; Dong et al., 2024b; Banerjee et al., 2025; Suresh et al., 2025; Firestone et al., 2025) languages, ensuring syntactic correctness. Recent grammar-constrained and type-constrained approaches rely on a prefix property (Mündler et al., 2025), where each partial program can be incrementally validated and completed into a syntactic and/or well-typed program. In contrast, REFINESTAT targets properties that cannot be prefix-checked, since statistical reliability requires full posterior inference rather than purely static validation. For more dynamic program generation, Poesia et al. (2022) and Ugare et al. (2025) implement error-driven backtracking.

Recently, several works have explored probabilistic inference/programming in LM-constrained generation. Loula et al. (2025) guide generation with potential scores and grammar rules, constraining token emission but not model correctness. In contrast, REFINESTAT generates probabilistic programs, enforces PPL checks during decoding, and retains only those passing Bayesian diagnostics. In Grand et al. (2025), a Planner writes an inference plan that LMs execute to satisfy constraints. REFINESTAT instead directly writes the probabilistic model and focuses on the quality of the posterior. Ahmed et al. (2025) adjusts next-token probabilities using a verifier so text matches high-level attributes. REFINESTAT aims for statistical validity, enforcing PPL semantics, and selecting the final program.

## 7 CONCLUSION AND LIMITATIONS

**Conclusion.** The main contribution of our work is to separate the task of generating probabilistic modeling through PPLs as fragments of priors and likelihood and to construct an LLM-based search procedure that automatically discovers the probabilistic program that satisfies the standard reliability metrics in the Bayesian workflow. We believe that our framework can be extended to enforce arbitrary reliability criteria defined by domain experts for reliable generation in other domains that involve domain-specific languages and plan to explore those in future work.

**Limitations.** While our framework incorporates key components of the Bayesian workflow (i.e., convergence diagnostics and predictive performance metrics), it does not include prior-predictive or posterior-predictive checks, which often require manual inspection and domain-specific judgment (Gelman et al., 1995). Thus, the reliability judgment is based on a subset of available diagnostics, and the reported ELPD only partially reflect model adequacy in some cases. Further, our refinement strategy is effective in practice but does not guarantee convergence to globally optimal program.

## 8 ACKNOWLEDGMENTS

This research was supported in part by NSF Grants No. CCF-1846354 and CCF-2313028.

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

# A  APPENDIX

This appendix provides expanded technical details, evaluations, and examples supplementing the main paper. Below is a structured index:

**A. Extended Background**

1. Language Models
2. Bayesian Workflow
3. Probabilistic Programming Language

**B. Probabilistic Program Metrics**

1. Basic Terminologies
2. Formal Definitions

**C. Example**

1. Illustrative Example
2. Examples of Structural Changes Introduced by REFINESTAT

**D. Prompt Design**

1. Prompt Template
2. Dataset-Specific Prompt

**E. Experimental Setup**

**F. Error Analysis**

1. Search Space Limitations
2. Model Misfit and Sampling Failures
3. Call-Level Hallucinations
4. Termination Failures Due to Budget Constraints

**G. Ablation Studies**

1. Token Efficiency Analysis
2. Memorization Effect

# A  EXTENDED BACKGROUND

## A.1  LANGUAGE MODELS

**Decoding and Constraints.**  Various approaches for token selection include greedy decoding, sampling, and beam search, repeated until an end-of-sequence (EOS) token or another stopping criterion is met. In constrained decoding, we may need to exclude specific tokens at certain positions. This is achieved using a mask $m \in \{0, 1\}^{|V|}$, where 1 indicates a viable token and 0 a discarded one. Decoding methods can then be applied to $m \odot softmax(\mathcal{S})$.

**Grammar-guided Generation**  Grammar-guided generation constrains model outputs to a formal grammar by using production rules of the form $A \rightarrow \alpha$, where $A$ is a nonterminal symbol and $\alpha$ is a sequence of nonterminals and *terminals* (the actual tokens or characters that appear in the final output). Most programming languages can be described using context-free grammar, with rules that apply to nonterminal symbols independent of their context. Grammar-guided generation ensures that LM outputs follow the syntactic structure required for, e.g., code generation or structured data formatting.

## A.2 Bayesian Workflow

At its core, a generative probabilistic program often follows a $\mathcal{D}\|\mathcal{P}\|\mathcal{L}$ structure: first fixing the observed data ($\mathcal{D}$), then sampling latent variables under the prior ($\mathcal{P}: p(z)$), and finally conditioning on the data via the likelihood ($\mathcal{L}: p(x \mid z)$).

Modern probabilistic programming languages such as Stan (Carpenter et al., 2017b) and PyMC (Salvatier et al., 2016) streamline this cycle by automating MCMC sampling and providing integrated diagnostics. These include prior predictive checks, convergence measures (e.g., split-$\widehat{R}$, effective sample size, BFMI, divergent NUTS transitions) (Vehtari et al., 2021; Hoffman et al., 2014; Betancourt, 2017), and predictive accuracy metrics such as PSIS-LOO (Vehtari et al., 2017).

These diagnostics guard against model mis-specification and poor sampling behavior, ensuring that only well-calibrated models are considered for inference or downstream decision-making.

## A.3 Probabilistic Programming Language

Probabilistic Programming Languages can be categorized as internal Domain-Specific Languages (DSLs), which embed within a host language and reuse its syntax and tooling (e.g., PyMC (Salvatier et al., 2016), NumPyro (Phan et al., 2019), Pyro (Bingham et al., 2019)), or as external DSLs that define their own syntax and compiler (e.g., Stan (Carpenter et al., 2017b)). This representation enables automated model specification while leveraging existing inference algorithms (Gordon et al., 2014).

# B    Probabilistic Program Metrics

## B.1 Basic Terminologies

Before presenting examples for individual diagnostics, we briefly define a few recurring terms that are used throughout:

**Chain:** An independent run of the sampler that generates a sequence of draws from the posterior distribution. Multiple chains are typically run to verify that results do not depend on initialization.

**Convergence:** The state in which all chains are sampling from the same region of the posterior distribution. Lack of convergence suggests that the sampler has not fully explored the posterior.

**Divergence:** A warning issued by the Hamiltonian Monte Carlo (HMC) algorithm indicating that numerical integration failed to follow the posterior geometry accurately. Divergences often signal problematic parameterizations or highly curved posterior regions.

## B.2 Formal Definitions

In our framework for valid statistical model synthesis, we employ several diagnostic metrics from probabilistic programming to ensure model validity. Below, we define each of these metrics formally.

**Definition 5 ($\widehat{R}$ Statistic)** *The split-$\widehat{R}$ statistic for parameter $\phi$, denoted $\widehat{R}\phi$, measures the convergence of Markov chains in MCMC sampling by comparing the between-chain variance to the within-chain variance. Formally:*

$$\widehat{R}\phi = \sqrt{\frac{V}{W}} \tag{1}$$

*where $V$ is the variance between chain means and $W$ is the average variance within chains. Values close to 1.0 indicate convergence, while higher values suggest poor mixing of chains.*

**Definition 6 (Effective Sample Size)** *The effective sample size (ESS) measures the equivalent number of independent samples obtained from autocorrelated MCMC draws. For parameter $\phi$, we*

*define:*

$$\text{ESS}_{\text{bulk},\phi} = \frac{MN}{\tau_{\text{bulk},\phi}} \tag{2}$$

$$\text{ESS}_{\text{tail},\phi} = \frac{MN}{\tau_{\text{tail},\phi}} \tag{3}$$

*where $M$ is the number of chains, $N$ is the number of draws per chain, and $\tau$ represents the autocorrelation time for bulk or tail estimates respectively. Bulk ESS evaluates sampling efficiency across the central mass of the posterior, while tail ESS focuses on the distribution tails.*

**Definition 7 (Divergent Transitions)** *Divergent transitions, denoted $\text{divergences}(M)$ for model $M$, count the number of leapfrog steps in Hamiltonian Monte Carlo where the numerical approximation of Hamiltonian dynamics breaks down due to extremely high curvature in the posterior geometry. These indicate potential pathological geometries in the posterior distribution that may lead to biased inference.*

**Definition 8 (Bayesian Fraction of Missing Information)** *The Bayesian Fraction of Missing Information, $\text{BFMI}(M)$ for model $M$, is defined as:*

$$\text{BFMI}(M) = \frac{\text{Var}(\Delta E)}{\text{Var}(E)} \tag{4}$$

*where $E$ represents the energy (negative log probability density) and $\Delta E$ is the change in energy between consecutive HMC iterations. Low BFMI values indicate poor exploration of the target distribution.*

**Definition 9 (Pareto Shape Parameter)** *The Pareto shape parameter $\widehat{k}_i(M)$ for observation $i$ in model $M$ quantifies the reliability of importance sampling estimates used in PSIS-LOO cross-validation:*

$$\widehat{k}_i(M) = \textit{shape parameter of Pareto distribution fitted to importance weights for observation } i \tag{5}$$

*Values $\widehat{k}_i < 0.5$ indicate reliable estimates, while $\widehat{k}_i > 0.7$ suggest unstable estimates that may require more robust computational approaches.*

**Definition 10 (Expected Log Pointwise Predictive Density)** *The Expected Log Pointwise Predictive Density under Leave-One-Out cross-validation, $\widehat{\text{elpd}}(M)$ for model $M$, measures the model's out-of-sample predictive accuracy:*

$$\widehat{\text{elpd}}(M) = \sum_{i=1}^{n} \log p(y_i \mid y_{-i}) \tag{6}$$

*where $p(y_i \mid y_{-i})$ is the predictive density for observation $i$ after fitting the model to all other observations. Higher values indicate better predictive performance.*

## C  EXAMPLES

### C.1  ILLUSTRATIVE EXAMPLE

We illustrate REFINESTAT on a standard Bayesian linear regression. Given a partial program $P$, we want to find a completion $M$ that maximizes the Bayesian reliability score $B(M)$. This example shows how each semantic and diagnostic check prunes or refines candidates.

1. **Partial Program $P$**
   At timestep $t$, REFINESTAT generates the following partial program:

   ```
   with pm.Model() as linear_model:
       alpha = pm.Normal("alpha", 0, 10)
       beta  = pm.Normal("beta", 0, 10)
       sigma = pm.HalfNormal("sigma", 5)
   ```

The LLM must complete the likelihood ($y_{\text{obs}}$).

2. **LLM Proposals & Semantic Checks**

   Candidate likelihoods are checked against PPL semantics as shown in Table 7:

Table 7: Semantic filtering of LLM-proposed likelihoods.

| Proposal | Check | Outcome |
|---|---|---|
| `y_obs = pm.ExtNormal("y_obs",` `mu=alpha + beta * x,` `sigma=sigma, observed=y)` | Distribution validity | **Reject** (hallucinated ExtNormal) |
| `y_obs = pm.Normal("y_obs",` `mu=alpha + beta * x,` `sd=sigma, observed=y)` | Parameter validity | **Reject** (deprecated sd vs. sigma) |
| `y_obs = pm.Normal("y_obs",` `mu=alpha + beta * x,` `sigma=sigma, observed=y)` | All checks | **Accept** |

3. **Diagnostic Checks & Guided Resampling** We run NUTS on the accepted model and observe:

   - $\widehat{R} = 1.2$ (too high),
   - 100 divergences.

   These failures reduce the Bayesian reliability score $B(M)$. REFINESTAT then resamples the likelihood or prior fragments (via the LLM) and retries inference until diagnostics ($\widehat{R}$, ESS, divergences, Pareto-$k$) fall within thresholds or the iteration limit is reached.

The final program $M^*$ converges ($\widehat{R} \approx 1$), shows zero divergences, and yields reliable ELPD-LOO. By pruning invalid distributions early and resampling based on diagnostic triggers, REFINESTAT iteratively refines candidates into a high-scoring $M^*$ with robust statistical reliability.

## C.2 EXAMPLES OF STRUCTURAL CHANGES INTRODUCED BY REFINESTAT

In addition to refining a fixed model skeleton, REFINESTAT can introduce qualitatively new structural components during the resampling process. When the model families and diagnostic feedback indicates low reliability, REFINESTAT may alter the functional form of the model, add new parameters, or change the dependency structure to explore alternatives with higher Bayesian reliability scores.

Below, we present three programs generated for the same dataset and initialization. These examples illustrate how REFINESTAT's refinement process can modify terms in the likelihood, expand the parameter space, or even change which variable is treated as observed.

Program 1: Standard linear regression

```
alpha = pm.Normal("alpha", mu=0, sigma=10)
beta  = pm.Normal("beta",  mu=0, sigma=10)
sigma = pm.HalfNormal("sigma", sigma=1)

mu = alpha + beta * year
likelihood = pm.Normal("likelihood", mu=mu, sigma=sigma, observed=C)
```

```
Program 2: Nonlinear likelihood and an additional prior

alpha = pm.Normal("alpha", mu=0, sigma=10)
beta  = pm.Normal("beta",  mu=0, sigma=10)
gamma = pm.Normal("gamma", mu=0, sigma=10)
sigma = pm.HalfNormal("sigma", sigma=10)

likelihood = (alpha + beta * year + gamma * year**2) * C + sigma * N
observed = pm.Normal("observed", mu=likelihood, sigma=sigma, observed=C)
```

```
Program 3: Changing the response variable and dependency structure

alpha = pm.Normal("alpha", mu=0, sigma=10)
beta  = pm.Normal("beta",  mu=0, sigma=10)
gamma = pm.Normal("gamma", mu=0, sigma=10)
sigma = pm.HalfNormal("sigma", sigma=10)

y_pred = alpha + beta * year + gamma * C
y_obs  = pm.Normal("y_obs", mu=y_pred, sigma=sigma, observed=N)
```

**Summary of Structural Differences.** Program 1 uses a standard linear regression and models `C` as a linear function of `year`. Program 2 expands the model by introducing a new parameter ($gamma$) and applying a nonlinear transformation involving $year^2$, `C`, and `N`, leading to a different generative process. Program 3 changes the response variable entirely, modeling `N` instead of `C`, and modifies the dependency graph by incorporating `C` as a covariate in the linear predictor.

These examples show that REFINESTAT is capable of exploring alternative model families and introducing new structure, such as additional priors, nonlinear link functions, or altered observational targets whenever such modifications remain semantically valid.

## D  PROMPT DESIGN

We use the same prompt across both the baseline, and REFINESTAT for experimentation purpose. To standardize the prompt across different datasets, we use a template in which the fields {description} and {template_code} are replaced with the dataset-specific description and code snippet, respectively.

```
Prompt Template

Template prompt:

# Complete the PyMC model definition within the 'with pm.Model() as m:' block below.
Your output must define a complete Bayesian model with appropriate priors, likelihood, and
then sample the posterior using, 'pm.sample(1000, tune = 1000, chains = 4,
return_inferencedata = True, idata_kwargs = {{"log_likelihood": True}})'. Do not
include any extra  commentary or text outside the code. Follow best practices for expert-level
Bayesian modeling.

# Description: {Description}

{Template_Code}
```

**Note:** The placeholders {description} and {template_code} are automatically substituted for each dataset. Below are the {description} and {template_code} respectively for each dataset:

## D.1 EIGHT SCHOOLS

**Description:** `A hierarchical model for the 8-schools data.`

```
Template Code

import pymc as pm
import numpy as np
import arviz as az
import matplotlib.pyplot as plt

# Data
y = np.array([28, 8, −3, 7, −1, 1, 18, 12])
sigma = np.array([15, 10, 16, 11, 9, 11, 10, 18])

with pm.Model() as m:
```

## D.2 DUGONGS

**Description:** `A growth model for dugongs with missing data.`

```
Template Code

import pymc as pm
import numpy as np
import arviz as az
import matplotlib.pyplot as plt

# Data
X = np.array([1, 1.5, 1.5, 1.5, 2.5, 4, 5, 5, 7, 8, 8.5, 9, 9.5, 9.5, 10, 12, 12, 13, 13, 14.5, 15.5,
15.5, 16.5, 17, 22.5, 29, 31.5])
y = np.array([1.8, 1.85, 1.87, 1.77, 2.02, 2.27, 2.15, 2.26, 2.47, 2.19, 2.26, 2.4, 2.39, 2.41, 2.5,
2.32, 2.32, 2.43, 2.47, 2.56, 2.65, 2.47, 2.64, 2.56, 2.7, 2.72, 2.57])

with pm.Model() as m:
```

## D.3 SURGICAL

**Description:** `The mortality rates in 12 hospitals performing cardiac surgery on babies.`

```
Template Code

import pymc as pm
import numpy as np
import arviz as az
import matplotlib.pyplot as plt

# Given Data
N = 12  # Number of observations
n = np.array([47, 148, 119, 810, 211, 196, 148, 215, 207, 97, 256, 360])
r = np.array([0, 18, 8, 46, 8, 13, 9, 31, 14, 8, 29, 24])

with pm.Model() as m:
```

## D.4 GP

**Description:** `Simulated data from a Poisson GP model.`

```
import pymc as pm
import numpy as np
import arviz as az
import matplotlib.pyplot as plt

# Given Data
N = 11  # Number of observations
x = np.array([−10, −8, −6, −4, −2, 0, 2, 4, 6, 8, 10])
y = np.array([4.75906, 1.59423, 2.99548, 5.27501, 1.66472, 2.24347, 2.8914, 4.08681,
4.60588, 0.802364, 3.92136])
k = np.array([40, 37, 29, 12, 4, 3, 9, 19, 77, 82, 33])

with pm.Model() as m:
```

## D.5 PEREGRINE

**Description:** `Simulated population counts of peregrines in the French Jura over 9 years`

```
import pymc as pm
import numpy as np
import arviz as az
import matplotlib.pyplot as plt

# Data

nyears = 40  # Number of years
year = np.array([−0.95, −0.9, −0.85, −0.8, −0.75, −0.7, −0.65, −0.6, −0.55, −0.5, −0.45, −0.4,
−0.35, −0.3, −0.25, −0.2, −0.15, −0.1, −0.05, 0, 0.05, 0.1, 0.15, 0.2, 0.25, 0.3, 0.35, 0.4, 0.45,
0.5, 0.55, 0.6, 0.65, 0.7, 0.75, 0.8, 0.85, 0.9, 0.95, 1])

C = np.array([27, 42, 35, 55, 61, 19, 41, 74, 43, 42, 73, 37, 48, 49, 19, 72, 30, 18, 31, 71, 63,
51, 48, 73, 49, 54, 43, 59, 30, 24, 62, 55, 51, 47, 14, 27, 45, 20, 26, 19])
N = np.array([43, 83, 53, 91, 95, 24, 62, 91, 64, 57, 97, 56, 74, 66, 28, 92, 40, 23, 46, 96, 91,
75, 71, 100, 72, 77, 64, 68, 43, 32, 97, 92, 75, 84, 22, 58, 81, 37, 45, 39])

with pm.Model() as m:
```

## E EXPERIMENTAL SETUP

We set the convergence threshold to $\alpha_R = 1.05$ for split-$\widehat{R}$, allow $\text{ESS}_{\text{bulk}} \geq \beta_{\text{bulk}} = 400$, and adopt a relaxed cutoff $\beta_{\text{tail}} = 100$ for $\text{ESS}_{\text{tail}}$ to accommodate lower sampling efficiency in the tails. For leave-one-out validation, $\widehat{\text{elpd}}(M)$ must be finite, with at least $1 - \epsilon = 0.8$ of data points having Pareto shape values below $L_{\text{cd}} = 0.7$. These thresholds are used consistently when computing the reliability score across all models. We use STANDARD unconstrained generation as our baseline.

Further, based on preliminary experiments we have chosen $\beta$ to be 4, $\alpha$ to be 2, and $R_{\max}$ as 100 for all experimental purposes.

We run experiments on a 48-core Intel Xeon Silver 4214R CPU with 2 NVidia RTX A5000 GPUs. REFINESTAT is implemented using PyTorch (Paszke et al., 2019), and Itergen library (Ugare et al., 2025) for refining the parser-guided LLM generation infrastructure. We run all experiments for 10 seeds to reduce result randomness, and use a temperature range of 0.2 to 0.4.

## F    ERROR ANALYSIS

### F.1    RUN RATE FAILURES

We categorized the failures by their root causes in different methods found during the run rate experiment:

- The Standard baseline exhibited frequent syntax errors (e.g., unmatched delimiters, missing imports) and invalid API calls.

- Syncode eliminated many basic syntactic mistakes but still suffered semantic errors, such as incorrect distribution parameter names, type mismatches, referring to deprecated API functions (e.g., calling `pm.sample_prior` from an earlier PyMC release), and inventing non-existent methods like `pm.random_coefs`.

- RefineStat, in contrast, often produced models whose samplers failed to explore the correct posterior modes, leading to chains stuck in low-density regions or divergent transitions, failures stemming from the model definitions rather than our decoding framework; REFINESTAT reduced both syntactic and semantic errors and avoided sampler pathologies by enforcing grammar and parameter validity during decoding.

### F.2    OTHER FAILURE MODES

While REFINESTAT significantly improves the syntactic correctness and statistical validity of generated probabilistic programs, we analyze the remaining failure cases to better understand the limitations of the framework and identify directions for future improvement.

**Search Space Limitations.**    Despite the use of resampling based mechanism for semantic validity, the language model occasionally reintroduces previously rejected code fragments. For instance, it may repeatedly generate outdated or invalid syntax such as the use of `sd` instead of `sigma` in PyMC model definitions:

```
mu = pm.Normal("mu", mu=0, sd=10)
```

This behavior likely stems from the model's prior exposure to deprecated APIs in its pretraining corpus and reflects the difficulty of escaping local attractors in the search space.

**Model Misfit and Sampling Failures.**    Despite generating semantically correct code, some models fail during posterior inference due to numerical instabilities inherent in the model specification. A common manifestation of this issue is the PyMC error:

```
SamplingError:  Initial evaluation of model at starting point
                            failed!
```

This error often arises when certain mathematical operations within the model, such as exponentiation or logarithms, result in undefined or non-finite values. For instance, exponentiating large numbers can lead to overflow, while taking the logarithm of zero or negative numbers is undefined. These numerical issues can cause the log-probability evaluations to return `NaN` or `inf`, thereby preventing the sampler from initializing properly.

**Call-Level Hallucinations.**    The model occasionally hallucinates invalid function names or API calls not present in the target probabilistic programming language. For example:

```
mu = pm.ExtNormal('ex', mu=0)
```

Such hallucinations highlight a mismatch between the syntactic plausibility and the executable validity of generated code, reinforcing the need for grounded semantic constraints during decoding.

**Termination Failures due to Budget Constraints.** In practice, we impose limits on the maximum number of iterations or generated tokens to maintain tractability. In some instances, these constraints are reached before a valid program is synthesized, resulting in truncated or incomplete code outputs.

**SamplingError** Initial evaluation of the model at the starting point failed due to numerical instabilities (overflow/NaNs).

**Indentation and commenting failures:** Wrong indentation of python-like function codes; inability to always close string comments.

## G ABLATION STUDY

### G.1 TOKEN EFFICIENCY ANALYSIS

To evaluate the computational cost associated with our framework, we measure the number of tokens consumed in generating a program under Itergen, Baseline (Unconstrained generation), REFINESTAT without Refinement Loop (REFINESTAT w/o RL), and REFINESTAT using Meta-LLama-3-8B. The token usage across five runs is recorded for each of these methods, from which we report the mean and standard deviation.

The Table 8 presents the results across models and datasets, with the final column ("Token Ratio") reporting the ratio of token usage by REFINESTAT relative to the baseline. On average, REFINESTAT consumes twice the number of tokens consumed by Baseline. While this reflects the added cost of our refinement mechanism, the overhead varies across settings. For instance, in some cases such as Dugongs, REFINESTAT uses fewer tokens than the baseline due to early convergence. Conversely, high multipliers (e.g., Eight Schools, GP) reflect continued refinement due to unmet stopping conditions, even if the generated program is already of high quality.

Table 8: Comparison of Itergen, Baseline, and REFINESTAT Variants with Multipliers.

| Dataset | Itergen | | Baseline | | REFINESTAT w/o RL | | REFINESTAT | | Token |
|---|---|---|---|---|---|---|---|---|---|
| | Mean | Std | Mean | Std | Mean | Std | Mean | Std | Ratio |
| Eight Schools | 98.8 | 5.3 | 606.8 | 386.3 | 147.6 | 36.7 | 1503.0 | 409.5 | 2.5x |
| Dugongs | 209.3 | 225.9 | 747.4 | 495.0 | 294.4 | 86.3 | 419.0 | 87.1 | 0.6x |
| GP | 131.3 | 8.9 | 663.6 | 410.9 | 155.0 | 21.2 | 1660.0 | 253.5 | 2.5x |
| Peregrine | 132.5 | 21.4 | 884.0 | 538.9 | 180.8 | 103.6 | 1740.0 | 418.0 | 2.0x |
| Surgical | 97.3 | 4.3 | 624.4 | 260.6 | 113.2 | 9.2 | 1275.0 | 1348.1 | 2.0x |
| | | | | | | | **Average Token Ratio** | | **1.9x** |

### G.2 MEMORIZATION EFFECT

To further stress-test our approach against memorization, we performed two controlled prompt modifications across all datasets using Meta-Llama 3-8B.

**Anonymized Prompt (REFINESTAT-AP):** All metadata and dataset names were removed, leaving only the raw dataset.

**Syntactic Obfuscation (REFINESTAT-SO):** All numerical values were transformed into exponential notation (e.g., 3.28e2 instead of 328) to prevent exact string matches with any potential training data.

**Anonymized Prompt and Syntactic Obfuscation (REFINESTAT-AP-SO):** Combined variant using both Anonymized Prompt, and Syntactic Obfuscation.

Table 9: Comparison of Diagnostic Scores and ELPD-LOO for REFINESTAT variants.

| Dataset | Variant | Reliab. Score ↑ | | $\widehat{R}$ ↓ | | ESS Bulk ↑ | | Divergences ↓ | | Pareto $k$ ↓ | | ELPD LOO ↑ | |
|---|---|---|---|---|---|---|---|---|---|---|---|---|---|
| | | Mean | Std | Mean | Std | Mean | Std | Mean | Std | Mean | Std | Mean | Std |
| Dugongs | REFINESTAT | 7.00 | 0.00 | 1.00 | 0.00 | 1696.00 | 284.33 | 0.00 | 0.00 | 0.00 | 0.02 | 8.42 | 24.51 |
| | REFINESTAT-AP | 7.00 | 0.00 | 1.00 | 0.00 | 1073.00 | 317.84 | 0.00 | 0.00 | 0.04 | 0.00 | -0.17 | 4.24 |
| | REFINESTAT-SO | 7.00 | 0.00 | 1.00 | 0.00 | 1753.50 | 68.95 | 0.00 | 0.00 | 0.04 | 0.00 | 8.31 | 0.09 |
| | REFINESTAT-AP-SO | 7.00 | 0.00 | 1.00 | 0.00 | 2257.00 | 731.23 | 0.00 | 0.00 | 0.00 | 0.00 | 1.79 | 7.09 |
| Eight Schools | REFINESTAT | 7.00 | 0.00 | 1.00 | 0.00 | 2303.00 | 768.76 | 0.00 | 0.00 | 0.00 | 0.05 | -31.77 | 0.61 |
| | REFINESTAT-AP | 7.00 | 0.00 | 1.00 | 0.00 | 2926.00 | 541.38 | 0.00 | 0.00 | 0.00 | 0.00 | -31.62 | 0.79 |
| | REFINESTAT-SO | 7.00 | 0.00 | 1.00 | 0.00 | 2470.50 | 32.61 | 0.00 | 0.00 | 0.06 | 0.06 | -31.61 | 0.01 |
| | REFINESTAT-AP-SO | 7.00 | 0.00 | 1.00 | 0.00 | 2187.50 | 252.83 | 0.00 | 0.00 | 0.00 | 0.00 | -31.60 | 0.04 |
| Peregrine | REFINESTAT | 7.00 | 0.00 | 1.00 | 0.00 | 3574.00 | 428.26 | 0.00 | 0.00 | 0.00 | 0.00 | -173.00 | 4.91 |
| | REFINESTAT-AP | 7.00 | 0.00 | 1.00 | 0.00 | 4057.00 | 811.85 | 0.00 | 0.00 | 0.00 | 0.00 | -173.14 | 65.91 |
| | REFINESTAT-SO | 7.00 | 0.00 | 1.00 | 0.00 | 1812.50 | 24.05 | 0.00 | 0.00 | 0.00 | 0.00 | -132.88 | 8.20 |
| | REFINESTAT-AP-SO | 6.00 | 0.00 | 1.00 | 0.00 | 1812.50 | 24.05 | 0.00 | 0.00 | 0.00 | 0.00 | -140.88 | 8.20 |
| GP | REFINESTAT | 6.00 | 0.49 | 1.00 | 0.00 | 1710.00 | 668.57 | 13.00 | 12.39 | 0.09 | 0.08 | -152.30 | 139.07 |
| | REFINESTAT-AP | 7.00 | 0.00 | 1.00 | 0.00 | 2283.00 | 519.55 | 0.00 | 0.00 | 0.00 | 0.04 | -21.24 | 2.58 |
| | REFINESTAT-SO | 7.00 | 0.00 | 1.00 | 0.00 | 1135.00 | 0.00 | 0.00 | 0.00 | 0.00 | 0.00 | -24.99 | 0.00 |
| | REFINESTAT-AP-SO | 6.50 | 0.53 | 1.00 | 0.00 | 1140.00 | 251.23 | 69.00 | 73.76 | 0.00 | 0.00 | -23.01 | 0.47 |
| Surgical | REFINESTAT | 6.00 | 0.77 | 1.00 | 0.00 | 579.00 | 1272.12 | 0.00 | 5.75 | 0.00 | 0.36 | -46.73 | 36.63 |
| | REFINESTAT-AP | 7.00 | 0.49 | 1.01 | 0.00 | 640.00 | 635.48 | 0.00 | 0.00 | 0.00 | 0.25 | -46.71 | 96.47 |
| | REFINESTAT-SO | 7.00 | 0.00 | 1.01 | 0.01 | 1311.00 | 742.99 | 0.00 | 0.00 | 0.04 | 0.04 | -65.22 | 19.85 |
| | REFINESTAT-AP-SO | 7.00 | 0.00 | 1.01 | 0.01 | 1139.00 | 638.22 | 0.00 | 0.00 | 0.04 | 0.04 | -69.27 | 24.20 |

As shown in Table 9, both variants match the original REFINESTAT on reliability, convergence, and predictive metrics. The table also reports a combined variant using both modifications, which performs comparably. This consistency suggests that REFINESTAT's effectiveness stems from learning from the provided data rather than memorization.

