# OpenReview forum: "RefineStat: Efficient Exploration for Probabilistic Program Synthesis"
_ICLR.cc/2026/Conference — ICLR 2026 Oral_

### Official Review · Reviewer_J73j · 2025-10-28

**Soundness:** 3
**Presentation:** 4
**Contribution:** 3
**Rating:** 8
**Confidence:** 3

**Summary:**

The paper introduces RefineStat a new framework for automatically synthesizing probabilistic programs using "smaller language models" (around 8B parameters). RefineStat introduces two checks for generating better probabilistic programs: 1. semantic and syntax checks to ensure the language models generates valid PyMC code (this is done using constrained decoding). 2. statistical diagnostic checks that ensure the results from posterior inference are reliable.

Empirically the paper presents results on synthesizing programs for some PosteriorDB datasets. RefineStat is evaluated using multiple open source language models as different base models.

Overall, I am in favor of accepting the paper because I think it provides a useful building block for setting up an automated workflow for synthesizing probabilistic programs. I do have some reservations about the empirical validation though which i would like the authors to address in the rebuttal.

**Strengths:**

The paper is very well written. The main arguments are easy to follow and the methodology is explained clearly. Both checks (semantic/syntax checks and statistical diagnostics) are sensible for any automated workflow in building Bayesian probabilistic models. I think the paper can serve as a useful stepping stone for building automated model building workflows.

**Weaknesses:**

There's a couple of weaknesses which I would like the authors to respond to in the rebuttal period. I don't think any of them should disqualify the paper from being accepted but I would appreciate some further discussion on the points listed below.

1. The paper motivates the use of small language models by the potential large API cost of using bigger language models. However, there is not a rigorous cost comparison of the RefineStat approach vs just calling the OpenAI-o3 model.  At least for the OpenAI model it would be good to provide the cost of the API calls.

2. Similarly, why is there no comparison with OpenAI-o3 in Table 1?

3. I appreciate the fact that the paper explicitly accounts for the fact that the empirical validation might be impacted by the fact that the PosteriorDB models could be contained in the training set of the language models. However, I find the empirical checks to be a bit unconvincing (although I am not an expert in the dataset memorization literature). Looking at the Table in Appendix G there is sometimes quite a lot of variation in the achieved scores (e.g. for Dugongs, GP and Surgical) and additionally, this check was only done on the LLama model whereas the evaluation in Table 3 uses the DQ-7B model as a base model. So the whole experimental setup is a bit inconsistent.

At the same time, dataset leakage is an unsolved problem as far as I am aware. So while this evaluation might be flawed, I think it shouldn't disqualify the paper from getting accepted.

4. The OpenAI-o3 model seems to be competitive with the RefineStat approach (on 8 schools, dugongs and surgical, if I am reading Table 3 correctly). So the big question is whether we really need RefineStat or whether training a bigger model will also improve probabilistic program synthesis performance. I don't see a reason though why RefineStat couldn't be used with a big model (other than cost) so it would have been nice to see some results for using OpenAI-o3 as a base model in RefineStat (not sure whether the constrained decoding is possible with a closed source model and/or it might be too expensive).

**Questions:**

L130: If this definition is taken from Vehtari et al, 2017 then the reference should be part of the definition.

Algorithm 1 misses any description for how $\mathcal{D}$, $\mathcal{P}$, and $\mathcal{L}$ are initially sampled.

Table 1: What does it mean for a value to be bold in the table?

Table 3: The table caption should state which metric is shown, I assume it is ELPD LOO? Additionally, could you show the standard deviation of the metrics (if I understand correctly these values are averages over 5 runs?)

---

> ### Author Response · Authors · 2025-11-27
>
> > why is there no comparison with OpenAI-o3 in Table 1?
>
> OpenAI o3 does not expose a user-controllable temperature parameter [refer: https://community.openai.com/t/o3-mini-unsupported-parameter-temperature/1140846/2]. Thus, its run rate cannot be evaluated across temperatures in a comparable way.
>
> > dataset leakage is an unsolved problem as far as I am aware. So while this evaluation might be flawed, I think it shouldn't disqualify the paper from getting accepted.
>
> We thank the reviewer for acknowledging our efforts to address one of the most pressing issues in the LLM ecosystem, dataset leakage and unintended memorization. We agree that our current memorization check mixes models, which can lead to confusion. Our intention was to provide a lightweight sanity check rather than a full leakage study, but we recognize that using a consistent model across experiments would strengthen the overall empirical setup. In the revision, we will standardize the model choice across the memorization check and the main results where feasible.
>
> > it would have been nice to see some results for using OpenAI-o3 as a base model in RefineStat
>
> Constrained decoding requires access to token-level logits at each decoding step to apply semantic masks. Closed-source models like OpenAI o3 do not expose these logits, so RefineStat cannot be implemented on top of them.
>
> > L130: If this definition is taken from Vehtari et al, 2017 then the reference should be part of the definition.
>
> > Algorithm 1 misses any description for how $\mathcal{D}$, $\mathcal{P}$, and \mathcal{L} are initially sampled
>
> Thank you for noting these details. We will update the text to include the reference directly within the definition and make Algorithm 1 more comprehensive to make it easier for the readers to understand.
>
> > Table 1: What does it mean for a value to be bold in the table?
>
> The bold values in Table 1 correspond to metrics where RefineStat outperforms the baseline. Since the reliability score depends on the underlying diagnostic metrics, we highlight it most often; when reliability scores are similar, we emphasize the ELPD values. In a few cases, we also bold metrics where the difference is particularly significant. We will make this more consistent and add a clarifying statement in the revised paper.
>
>
> > Table 3: The table caption should state which metric is shown, I assume it is ELPD LOO? Additionally, could you show the standard deviation of the metrics (if I understand correctly these values are averages over 5 runs?)
>
> We thank the reviewer for pointing this out; indeed, it is ELPD LOO. We didn’t explicitly include the standard deviation in this table, simply because prior work like BoxLM doesn’t include the standard deviation corresponding to these datasets, and doesn’t provide the code as well to repeat their experiments. For completeness, we now report the mean ± standard deviation for RefineStat w/ DQ-7B and OpenAI-o3 below:
>
> | Dataset       | Expert   | BoxLM w/ GPT-4 | REFINESTAT w/ DQ-7B (mean ± std) | OpenAI-o3 (mean ± std) |
> |---------------|----------|----------------|-----------------------------------|-------------------------|
> | Eight schools | -30.70   | -30.42         | -30.68 ± 0.11                    | -30.74 ± 0.07           |
> | Dugongs       | 22.43    | 23.40          | 8.35 ± 0.06                      | 22.83 ± 8.12            |
> | Peregrine     | -112.60  | -173.11        | -114.29 ± 2.76                   | -133.29 ± 10.33         |
> | Surgical      | -39.73   | -38.03         | -46.51 ± 3.49                    | -38.73 ± 0.51           |
> | GP            | -26.53   | –              | -23.39 ± 1.14                    | -34.95 ± 13.28          |
>
>
> Note that, unlike RefineStat, the OpenAI-o3 results do not apply any reliability-score based filtering of programs.

---

### Official Review · Reviewer_uUBR · 2025-10-29

**Soundness:** 3
**Presentation:** 3
**Contribution:** 3
**Rating:** 6
**Confidence:** 3

**Summary:**

The paper proposes REFINESTAT, a probabilistic program synthesis framework that allows small open-weight language models to automatically generate Bayesian models that are both executable and statistically trustworthy.

REFINESTAT has two core components:

- Semantically constrained generation.
Instead of allowing an LLM to free-generate PyMC code, REFINESTAT enforces a set of domain-specific validity predicates at generation time. These constraints check: parseability, valid distribution names, correct parameter names/signatures, shape/type compatibility, support (e.g. positive-only scales), and dependency ordering between random variables. If a generated fragment violates one of these constraints, that fragment alone is locally rejected and resampled, rather than discarding the entire program. This prunes away statistically nonsensical or sampler-hostile code before it ever reaches inference.

- Diagnostic-aware refinement.
For each candidate model, REFINESTAT runs inference in PyMC (using NUTS/HMC) and computes standard Bayesian workflow diagnostics. These are aggregated into a Bayesian Workflow Reliability Score, which counts the number of diagnostics that pass. If the score is below a threshold, REFINESTAT selectively resamples just the likelihood block or just the prior block, keeping the rest of the model fixed, and tries again. A model is accepted only if it reaches sufficient reliability and good predictive performance.


The framework is tested on five canonical PosteriorDB problems (Eight Schools, Dugongs, Surgical, Peregrine, Poisson GP) using several open models and compared to:

- naive prompting (unconstrained generation),
- syncode (syntax-only constrained decoding),
- multi-agent GPT-4 systems such as BoxLM,
- expert-written Stan models,
- and proprietary baselines like OpenAI o3.

**Strengths:**

Prior constrained decoding work in LLM code generation typically enforces syntactic validity or DSL grammar. REFINESTAT goes further and embeds statistical semantics: valid distribution calls, correct parameter names and shapes, valid supports (e.g., positive scale parameters), and dependency ordering between latent variables and likelihood terms.

Instead of treating generation as one-shot code synthesis, REFINESTAT adopts the classical Bayesian workflow loop: propose a model $\rightarrow$ run inference $\rightarrow$ diagnose $\rightarrow$ revise. It encodes this loop in a way that is easily enforceable by turning convergence/efficiency diagnostics.

Competing systems like BoxLM rely on multi-agent GPT-4 setups (generator + critic). REFINESTAT shows that structured guidance, combined with principled diagnostics, can enable a single, small open model to match or beat these heavier pipelines on some benchmarks.

REFINESTAT increases run rate from $\approx$10–11% (unconstrained)/ $\approx$21% (syntax-only) to $\approx$45–50%. Ablation studies show that removing specific semantic checks (especially parameter validity) sharply degrades success, establishing that each constraint is important.

A weaker open model (DeepSeek-R1-Distil-Qwen-7B), which fails with naive prompting, becomes capable of producing usable models on all datasets when run through REFINESTAT.

**Weaknesses:**

All experiments are on five classic PosteriorDB tasks (Eight Schools, Dugongs, Surgical, Peregrine, Poisson GP). These are relatively low-dimensional, reasonably clean, and heavily studied.
The author doesn’t use any messy, modern, high-dimensional problems:
- hierarchical GLMs with a large number of predictors,
- data with missing values or covariate drift,
- settings where NUTS is brittle.

The paper claims progress toward automated statistical model discovery, but current evidence is confined to highly curated, small-scale Bayesian exemplars.

All experiments are in PyMC. The authors do not demonstrate portability. The semantic predicates (distribution names, argument schemas, support typing, etc.) are currently PyMC-specific.

The refinement loop fixes models by resampling either the likelihood or the prior block when diagnostics fail, and continues iterating until it passes a threshold. This is a clever local repair heuristic, but it also has a limitation: it assumes the proper global structure (hierarchy, link function, GP terms, etc.) was already present in the initial model draft.

The paper has to quantify how often the system gets stuck in such local minima (tinkering with scale priors on a fundamentally wrong structure) or how frequently it must abandon a structure entirely.

The thresholds are hand-picked. The paper does not provide a sensitivity analysis to show robustness.

The method repeatedly runs full NUTS/HMC inside the loop to evaluate diagnostics and to drive refinement. That’s the expensive part.
The paper does discuss token usage (and claims REFINESTAT avoids GPT-4-scale inference costs by using only small open models). Still, it does not report: (i) the number of complete inference runs per dataset per accepted model, and (ii) wall-clock time.

**Questions:**

During refinement, can REFINESTAT introduce qualitatively new structure (e.g., add a GP term, change link functions, etc.), or does it mostly resample priors/likelihoods within the same initial skeleton? Empirically, how often does refinement change the model family, versus just adjusting hyperparameters? If it does not change structure mid-loop, do you consider that a limitation or is it a deliberate design choice?

When diagnostics fail, what feedback does the LLM get?

Are the thresholds fixed globally across all datasets/LLMs, or tuned per task?

Can you quantify average resource requirements? How many complete HMC/NUTS inference runs does REFINESTAT perform per dataset, on average, before it finds an acceptable model? What’s the typical wall-clock time order of magnitude per dataset on your hardware (minutes, hours)?

For a moderately larger dataset, is the loop still tractable, or does it become prohibitively slow?

It would be helpful to understand how naive prompting + the same diagnostic-aware refinement loop, but without semantic constraints, does. This ablation would isolate the marginal benefit of semantic-constrained decoding and strengthen the argument that both components are essential.

---

> ### Author Response · Authors · 2025-11-27
>
> > During refinement, can REFINESTAT introduce qualitatively new structure (e.g., add a GP term, change link functions, etc.), or does it mostly resample priors/likelihoods within the same initial skeleton?
>
> > Empirically, how often does refinement change the model family, versus just adjusting hyperparameters?
>
> Yes, REFINESTAT does introduce new structure during the resampling process which enables it to choose the best possible program among the wide variety of programs it explores during resampling. Below, we present example programs that illustrate how the structure of the generated programs can change.
>
> The model family or hyperparameter change varies from dataset to dataset and from seed to seed, based on the constraints that we have defined. To analyze the change in the model family, and adjustments in hyperparameter, we analyzed one particular dataset which in this case was GLM. As compared to a single base program, 60% of them had different likelihood functions, and the remaining program had different hyperparameters in prior. Example programs are as shown below:
>
> Program 1:
>
>     # Define priors for the parameters
>     alpha = pm.Normal("alpha", mu=0, sigma=10)
>     beta = pm.Normal("beta", mu=0, sigma=10)
>     sigma = pm.HalfNormal("sigma", sigma=1)
>
>     # Define the model
>     mu = alpha + beta * year
>
>     # Define the likelihood
>     likelihood = pm.Normal("likelihood", mu=mu, sigma=sigma, observed=C)
>
>
> Program 2:
>
>     # Define priors for the parameters
>     alpha = pm.Normal("alpha", mu=0, sigma=10)
>     beta = pm.Normal("beta", mu=0, sigma=10)
>     gamma = pm.Normal("gamma", mu=0, sigma=10)
>     sigma = pm.HalfNormal("sigma", sigma=10)
>
>     # Define the likelihood
>     likelihood = (alpha + beta * year + gamma * year**2) * C + sigma * N
>
>     # Define the observed data
>     observed = pm.Normal("observed", mu=likelihood, sigma=sigma, observed=C)
>
>
> Program 3:
>
>     # Define priors for the parameters
>     alpha = pm.Normal("alpha", mu=0, sigma=10)
>     beta = pm.Normal("beta", mu=0, sigma=10)
>     gamma = pm.Normal("gamma", mu=0, sigma=10)
>     sigma = pm.HalfNormal("sigma", sigma=10)
>
>     # Define the likelihood
>     y_pred = alpha + beta * year + gamma * C
>     y_obs = pm.Normal("y_obs", mu=y_pred, sigma=sigma, observed=N)
>
> To clarify the structural differences between the three programs shown above, we summarize their key distinctions below:
>
> Program 1 uses a simple linear regression with priors on alpha, beta, and sigma, and models C directly as a linear function of year.
>
>
> Program 2 introduces an additional prior on gamma and changes the likelihood to a nonlinear form (alpha + beta * year + gamma * year**2) * C + sigma * N, resulting in a different generative process.
>
>
> Program 3 also adds the extra prior on gamma but changes the response variable entirely, modeling N instead of C using a linear function of both year and C, which alters the dependency structure of the model.
>
>
> > When diagnostics fail, what feedback does the LLM get?
>
> At present, we are simply resampling the prior or the likelihood based on a heuristic-based approach. We plan to incorporate many more feedback-driven diagnostics, like prior predictive check and posterior predictive checks, to specifically get the feedback corresponding to either the prior or posterior.
>
> > Are the thresholds fixed globally across all datasets/LLMs, or tuned per task?
>
> The thresholds are fixed globally across all datasets/LLMs to avoid any unfair advantage that we may provide based on the different hyperparameter settings specific to the dataset/LLM. Further, this choice was considered so that we can do a fair comparison to BoxLM, which also keeps the hyperparameter consistent across all datasets/LLMs.
>
> > Can you quantify average resource requirements? How many complete HMC/NUTS inference runs does REFINESTAT perform per dataset, on average, before it finds an acceptable model? What’s the typical wall-clock time order of magnitude per dataset on your hardware (minutes, hours)?
>
> Total number of complete HMC/NUTS inference runs REFINESTAT performs per dataset are as follows (averaged across different LLMs):
> Dugongs: 3.5, Eight Schools: 6.5, GLM: 2.25, GP: 3.5, Surgical: 3
>
> The implementation hasn’t been optimized for performance yet, but a usual run for the datasets that we have used takes about 5 minutes on average
>
> > For a moderately larger dataset, is the loop still tractable, or does it become prohibitively slow?
>
> Yes, the loop still remains tractable despite using larger or different sized datasets, primarily because the method's runtime depends on how quickly the Small Language Model discovers the first 4 successfully running programs and whether the generated programs have errors or not.

---

> > ### Author Response · Authors · 2025-11-27
> >
> > > It would be helpful to understand how naive prompting + the same diagnostic-aware refinement loop, but without semantic constraints, does. This ablation would isolate the marginal benefit of semantic-constrained decoding and strengthen the argument that both components are essential.
> >
> > We agree that this ablation is important. A similar “naive prompting + feedback loop” strategy has been explored previously in BoxLM [Li et al., ICML 2024] for automated statistical modeling, though their experiments relied on GPT-4 models.
> >
> > When we replicated this setup with Small Language Models, we found that it performs poorly: the limited context window leads to frequent hallucinations and, critically, the refinement feedback is often not incorporated into subsequent generations. As a result, the refinement loop fails to reliably improve model quality. This observation is precisely why we introduce semantic-constrained decoding. Without it, SLMs do not consistently respect feedback or maintain structural correctness across iterations.

---

> ### Comment · Reviewer_uUBR · 2025-11-28
>
> Thanks for your detailed response; most of my doubts have been answered, and I am positive about the paper.

---

### Official Review · Reviewer_oWB5 · 2025-10-31

**Soundness:** 3
**Presentation:** 3
**Contribution:** 3
**Rating:** 4
**Confidence:** 4

**Summary:**

The paper presents a method for generating a probabilistic program using a "small" LLM, where during generation process certain semantic checks are enforced. This method can potentially improve the reliability of the final output. A variety of techniques are currently being proposed to check correctness at decoding time,  this paper differs due to its focus on probabilistic models and their properties. The method is evaluated on a  set of benchmarks against baselines including unconstrained generation by powerful LLMs as well as prior syntax-constrained generation methods.

**Strengths:**

The problem is well motivated and the basic approach is an appealing way to address the challenges.

**Weaknesses:**

The authors cite a number of papers proposing constrained decoding for classical programs, but a meaningful comparison is missing. The technique, as a result, seems incremental. The experimental results are limited and not particularly impressive.

**Questions:**

1. Can you explain the key differences between this and constrained decoding (e.g. "Type-Constrained Code Generation with Language Models" in PLDI 2025). Are there new challenges or just adaptation to probabilistic settings.
2. Can you explain how synthesis algorithms works using an illustrative example?
3. Based on the experimental results, the method doesn't seem "ready" for real-world applications. That's ok, but can you articulate what advances will be needed for this?

---

> ### Author Response · Authors · 2025-11-21
>
> We thank the reviewer for their constructive feedback.
>
> > The authors cite a number of papers proposing constrained decoding for classical programs, but a meaningful comparison is missing.
>
> > Can you explain the key differences between this and constrained decoding (e.g. "Type-Constrained Code Generation with Language Models" in PLDI 2025). Are there new challenges or just adaptation to probabilistic settings?
>
> The primary distinction is that prior grammar-constrained and type-constrained generation techniques enforce a prefix property (see pg 17, Definition 2 in the PLDI’25 paper), where each program prefix can be validated incrementally, such that it can be completed into a syntactic or a well-typed program. In contrast, REFINESTAT addresses properties of probabilistic programs that cannot be reduced to an equivalent prefix property. Statistical reliability measured through convergence diagnostics, divergences, BFMI, and PSIS-LOO can only be assessed after running full posterior inference on a complete program.
> Our diagnostic-aware refinement uses this Bayesian workflow feedback to iteratively improve models until they are statistically reliable, not just satisfying static semantics. Besides ensuring syntactic validity by building on grammar-constrained generation, our core contribution includes enabling efficient exploration of the space of statistically sound probabilistic programs through post-inference diagnostic guidance, which is not supported in prior works such as Mundler et. al 2025.
>
>
> We will make this comparison with PLDI’25 and similar works more explicit in the paper.
>
> > Can you explain how synthesis algorithms work using an illustrative example?
>
> We provide a detailed example in Appendix C (Page 16). The example shows how our semantic checks filter invalid distributions and parameters, followed by diagnostic-based refinement when convergence fails. We will explicitly point to this example in the main text and expand it further if helpful.
>
>
> > Based on the experimental results, the method doesn't seem "ready" for real-world applications. That's ok, but can you articulate what advances will be needed for this?
>
> Our current results on standard benchmarks show the practicality of our approach. To further improve the real-world applicability of our work, we believe that the following research directions are promising.
> 1. Extending evaluation to larger datasets with missing data, high-dimensional problems, and diverse modeling scenarios
> 2. Incorporating fine-grained feedback from the diagnostics loop for better resampling of specific portions of the probabilistic program
>
> We will include this discussion in the next revision of the paper.

---

> > ### Comment · Reviewer_oWB5 · 2025-11-24
> >
> > Thanks for the response. Based on authors' response and the other reviews, I have decided to raise my score.

---

### Official Review · Reviewer_PPbt · 2025-11-01

**Soundness:** 3
**Presentation:** 3
**Contribution:** 3
**Rating:** 8
**Confidence:** 5

**Summary:**

The submission proposes a framework for constrained generation of probabilistic programs using a small language model and is based on both static and inference-based statistical checks to resample and improve the program. The framework is evaluated on several language model using a set of public datasets from statistical literature and shows solid performance.

The claimed contributions are

* a method for SLM based generation of probabilistic programs ensuring synatcic and semantic correctness and predictive performance

* the use of constrained decoding for generation of probabilistic programs

* iterative resampling of probabilistic programs using an unmodified small language model

* an evaluation baseline of probabilistic program generation based on a number of criteria outlining syntactic, semantic, and statistical correctness.

**Strengths:**

The submission presents a non-trivial view on incorproration of Bayesian workflow into Large/Small language model pipeline. It is good to see Bayesian workflow to be formalized and automated for data-driven probabilistic program search. The method is described in detail, supported by clean source code to reproduce experiments, and thoroughly evaluated.

**Weaknesses:**

Major:

1. I am concerned about evaluating any language model based method on datasets broadly represented in the statistical literature. While the authors argue that through ablation studies they demonstrates that the strengths of RefineStat come from learning rather from memoization, independent empirical evidence of this does not seem to be very convincing.

2. I am wondering whether PyMC is special in any way to specifically support the approach, how easy to apply the approach to other inference frameworks (Stan, Pyro/NumPyro, DynamicHML.jl, Turing.jl etc). I do not think PyMC is  sufficiently justified in the paper as the 'natural' choice.

Minor: while the authors make a very respectable effort to define everything, the clarity of definitions and notation throughout the paper can be improved. For example:

Lines 130-132 define PSIS-LOO in terms of Pareto smoothed ways. Lines 134-136 that follow mention the fitted Pareto shape parameters k_i, not defined previously. Either more details should be provided about Pareto smoothing in the definion, or fewer details in what follows. As it is now, the reader stumbles on this and unless happens to be familiar with pareto-smoothed cross-validation, has to read the cited paper to understand what is going on.

Lines 173-180 define the score as the sum of 7 'binary indicators'. The 'indicators' are defined as booleans rather than indicators, and booleans cannot be summed up in mathematics (they can in Python). Some sound notation should be used so that s_j is 1 if the condition holds and zero otherwise. Then, in lines 187-192 are quite confusing, at least on the first reading, giving an example when elpd cannot be computed and then immediately saying that the higher the score is, the more the value of elpd can be trusted.

Lines 220-223 talk about how std= is invalid and RefineStat correctly replace std with sigma, but in the accompanying Figure 2 there is sd= rather than std=.

Pedro Domingos appears as Domingos in line 440 and as Pedro in lin 442 in citation anchors.

These are just a few examples, there are quite some more such glitches, and this impairs the reader's ability to understand the paper. The paper should be proofread and fixed.

**Questions:**

How easy/challenging is it to retarget RefineState to Stan?

---

> ### Author Response · Authors · 2025-11-27
>
> We thank the reviewer for the feedback and presentation issues, which we will fix.
>
> > While the authors argue that through ablation studies they demonstrate that the strengths of RefineStat come from learning rather from memoization, independent empirical evidence of this does not seem to be very convincing.
>
> We acknowledge that contamination is a broader challenge faced by virtually all LLM evaluations, and our setting is not unique in this regard. While certain domains have begun addressing this through live evaluation frameworks (e.g., LiveCodeBench), developing an analogous live benchmark for probabilistic programming would be a valuable direction for future work. Although such an evaluation infrastructure does not yet exist, we recognize this limitation and have mitigated it to the best of our ability through the additional experiment included in the paper. We will also make this limitation explicit in the revised version of the paper.
>
> > I am wondering whether PyMC is special in any way to specifically support the approach, how easy to apply the approach to other inference frameworks (Stan, Pyro/NumPyro, DynamicHML.jl, Turing.jl, etc).
>
> We choose PyMC as the DSL for implementing REFINESTAT so that we can do an appropriate comparison with the existing work in this field, BoxLM[Li et al., ICML 2024], which solely used PyMC in its implementation. REFINESTAT can be applied across all different inference frameworks, but since constrained generation is a major component of REFINESTAT, a crucial component is writing appropriate grammar for the language of choice. Hence, with our current grammar, we can expand REFINESTAT to other Python-based PPLs like PyMC, Numpyro, etc.
>
> We conducted experiments using REFINESTAT for different temperatures using Codegemma-7B, and Qwen-2.5-Coder on Numpyro, and observed that using our framework, we are able to improve the Run Rates of the program by a factor of 2x as compared to standard unconstrained generation. The compilation-rate, and the run-rate results are as shown below.
> Run Rate:
> | Temp. | Standard | RefineStat |
> |-------|----------|-------------|
> | 0.2| 0.17| **0.34**|
> | 0.3| 0.15| **0.35**|
> | 0.4| 0.15| **0.33**|
>
> Further, similar to Table 2, we present the results for Qwen-2.5-Coder across all datasets in the case of NumPyro, as shown below. The results clearly indicate that RefineStat outperforms Standard unconstrained generation in terms of the Reliability Score; and when the Reliability Scores are equal, RefineStat also achieves better ELPD-LOO scores.
>
> | Dataset      | Variant     | Reliab. Sc ↑ (Mean) | Reliab. Sc ↑ (Std) | R̂ ↓ (Mean) | R̂ ↓ (Std) | ESS Bulk ↑ (Mean) | ESS Bulk ↑ (Std) | Diverg. ↓ (Mean) | Diverg. ↓ (Std) | Pareto k ↓ (Mean) | Pareto k ↓ (Std) | ELPD LOO ↑ (Mean) | ELPD LOO ↑ (Std) |
> |--------------|-------------|----------------------|----------------------|-------------|------------|---------------------|-------------------|-------------------|------------------|---------------------|-------------------|--------------------|-------------------|
> | **GP** | Standard| × | × | × | × | × | × | × | × | × | × | × | × |
> |              | RefineStat | 6.67 | 0.47 | 1.00 | 0.00 | 2344.33 | 115.46 | 0.00 | 0.00 | 0.09 | 0.13 | -83.07 | 82.70 |
> | **Surgical**| Standard| 6.00 | 0.00 | 1.00 | 0.00 | 1472.00 | 0.00 | 0.00 | 0.00 | 0.25 | 0.00 | -245.14 | 0.00 |
> |              | RefineStat | 7.00 | 0.00 | 1.00 | 0.00 | 1628.00 | 0.00 | 0.00 | 0.00 | 0.08 | 0.00 | -50.37 | 0.00 |
> | **Peregrine** | Standard   | 6.00 | 0.00 | 1.00 | 0.00 | 3232.50 | 81.32 | 0.00 | 0.00 | 0.00 | 0.00 | -368.21 | 0.04 |
> |              | RefineStat | 6.67 | 0.47 | 1.00 | 0.00 | 1658.00 | 135.60 | 0.00 | 0.00 | 0.11 | 0.10 | -76.25 | 122.44 |
> | **8 Schools** | Standard   | 7.00 | 0.00 | 1.00 | 0.00 | 1769.00 | 0.00 | 0.00 | 0.00 | 0.13 | 0.00 | -31.83 | 0.00 |
> |              | RefineStat | 7.00 | 0.00 | 1.00 | 0.00 | 1850.00 | 50.00 | 0.00 | 0.00 | 0.11 | 0.02 | -31.10 | 0.05 |
> | **Dugongs**   | Standard   | 7.00 | 0.00 | 1.00 | 0.00 | 1739.00 | 0.00 | 0.00 | 0.00 | 0.04 | 0.00 | 8.18 | 0.00 |
> |              | RefineStat | 7.00 | 0.00 | 1.00 | 0.00 | 1747.00 | 11.55 | 0.00 | 0.00 | 0.04 | 0.00 | 8.19 | 0.01 |
>
> None of the programs generated under Standard Unconstrained decoding work for the GP setting, as they fail to include a likelihood term. In contrast, RefineStat’s resampling mechanism produces valid programs and yields appropriate results. Further, RefineStat achieves higher Reliability Scores for both Surgical and Peregrine.
>
> > How easy/challenging is it to retarget RefineState to Stan?
>
> Since Stan is not a Python-based language, we need to write an appropriate Lark (LR) grammar that follows all the conventions defined in Stan.
>
> [1] Li et al., Automated Statistical Model Discovery with Language Models, ICML 2024
>
> [2] Jain, Naman, et al. "Livecodebench: Holistic and contamination free evaluation of large language models for code." arXiv preprint arXiv:2403.07974 (2024).

---

### Author Response · Authors · 2025-12-03

We thank all reviewers for their constructive and positive engagement. Reviewers consistently noted that the clarifications and expanded experiments strengthened the submission and resolved their major concerns. Below, we summarize each reviewer’s key point and the corresponding update in the revised paper.

**Reviewer PPbt** asked whether RefineStat is tied to PyMC or can be applied to other probabilistic programming systems, and also requested clearer diagnostic definitions.
**Update:** We revised Sections 2–3 to clarify PSIS-LOO, Pareto-k, and the reliability score with correct indicator notation. To directly address adaptability, we added Section 5.5, which provides new NumPyro experiments showing that RefineStat significantly improves run rate and diagnostic reliability across all datasets, demonstrating that the framework generalizes beyond PyMC and is not tied to a single PPL implementation.

**Reviewer oWB5** asked for a clearer distinction between our method and classical constrained decoding techniques like “Type-Constrained Code Generation with Language Models”.
**Update:** We expanded the discussion in Section 6 to explain why Bayesian-diagnostic constraints cannot be enforced via prefix validity, distinguishing our approach from existing works.

**Reviewer uUBR** asked whether refinement can alter model structure rather than only hyperparameters.
**Update:** We expanded Appendix C to include concrete program examples and an explanation of how refinement changes model components. These examples illustrate how RefineStat can switch likelihood families, introduce new priors, and alter dependency structures between variables, demonstrating that the refinement mechanism supports meaningful structural evolution.

**Reviewer J73j** requested clearer explanation of Algorithm 1, better clarification of how to read Table 2, and complete reporting (including missing standard deviations) in the table comparing OpenAI-o3 with RefineStat.
**Update:** We refined the description of the initial sampling of program components around Algorithm 1 in Section 3.2. We also updated the discussion of Table 2 in Section 5.2 to more clearly explain how the bolded values should be interpreted. Finally, we revised the OpenAI-o3 vs. RefineStat comparison table in Section 5.3 to include mean ± std ELPD-LOO values.

The discussion consensus was strongly positive: two reviewers assigned high scores, one reviewer explicitly increased their score, and another stated they were positive about the paper after reading the rebuttal. We believe the revised manuscript fully addresses the reviewers’ concerns.

---

### Meta-Review · Area_Chair_nWbv · 2025-12-21

**Summary:**

The authors introduce an automation pipeline based on the Bayesian workflow and (small) LLMs for probabilistic programs. For this to produce valid and "reliable" programs, the authors perform semantic constrained decoding of the program and an iterative refinement loop, that is inspired by the Bayesian workflow, to find a program definition which results in reliable posterior estimates (e.g., high ESS). The method is evaluated for different choices of LLM backends on a selection of datasets from posteriorDB indicating that their novel combination is successful.

**Reviewer Concerns:**

Most of the reviewers' concerns have been resolved during the rebuttal, with several improvements of the paper done or promised by the authors. Some concerns, however, remain. Specifically, reviewers have been concerned about contermination in the evaluation as the posteriorDB datasets are fairly standard and potentially indexed by recent LLMs. The concern was acknowledged by the authors, and I believe a more detailed investigation of potential contamination is needed.

**Reviewer Scores:**

Reviewers have been very positive about this submission. Here are potential outcomes after the rebuttal phase.

- PPbt: The reviewer has been very positive and in support of the submission and will likely keep the initial score.
- oWB5: The reviewer mentioned having increased the score (e.g., to a 5/6).
- uUBR: The reviewer expressed satisfaction after the rebuttal and likely would have increased the initial score (e.g., to a 7/8).
- J73j: The reviewer has been very positive about the submission and will likely keep the initial score.

---

### Decision · Program_Chairs · 2026-01-26

Accept (Oral)